# Reinforcing Multi-Turn Reasoning in LLM Agents via Turn-Level Reward Design

## Abstract

This paper investigates Reinforcement Learning (RL) approaches to enhance the reasoning capabilities of Large Language Model (LLM) agents in long-horizon, multi-turn scenarios. Such multi-turn agentic tasks can be naturally formalized as turn-level Markov Decision Processes (MDPs). However, most existing methods adopt MDP formulations with trajectory-level rewards, either terminal rewards that provide only a final outcome signal, or delayed rewards that merge intermediate and outcome signals into a single sparse feedback, leading to poor credit assignment. To address this limitation, we reformulate these tasks as MDPs with explicit turn-level rewards and provide theoretical analysis supporting the effectiveness of this design. Building on this formulation, we extend popular RL algorithms, GRPO and PPO, to their respective multi-turn variants, enabling fine-grained credit assignment. We conduct case studies on multi-turn reasoning-augmented search agents, where we carefully design two types of turn-level rewards: verifiable and LLM-as-judge. Our experiments on multi-turn search tasks demonstrate that our proposed formulation, incorporated well-designed turn-level rewards, enables RL algorithms to significantly outperform baseline methods with trajectory-level rewards. Both training and validation reward curves illustrate that our method achieves *greater stability*, *faster convergence*, and *higher accuracy*. Numerical results across diverse question-answering datasets further show that our approach consistently delivers highest answer correctness and 100% format correctness.

## 1 Introduction

Reinforcement Learning (RL) has recently emerged as a powerful approach for improving the reasoning capabilities of Large Language Models (LLMs), allowing them to explore and refine long Chains of Thought (CoT) (Wei et al., 2022) in complex decision-making tasks. Building on this paradigm, reasoning-based LLMs, such as OpenAI's o1 (Jaech et al., 2024) and DeepSeek's R1 (Guo et al., 2025a), demonstrate remarkable performance in textual reasoning tasks by learning analytical thinking and self-reflection. Despite these advancements, LLMs that rely solely on textual reasoning remain limited in tasks that require precise and complex numerical computation, information retrieval from web pages or local databases, or code execution. Equipping LLMs as autonomous agents with access to external tools, such as search engines, scientific calculators, or code interpreters, can significantly extend their capabilities beyond pure text-based reasoning (Gou et al., 2023).

Training LLMs to operate as autonomous agents in interactive environments faces unique challenges. Agent settings often require models to make sequential, multi-turn decisions in complex reasoning tasks. Many existing approaches (Chen et al., 2025b; Jin et al., 2025b; Feng et al., 2025a) formulate these multi-turn interactive tasks as single-turn problems, relying solely on final outcome-level rewards such as answer correctness. Popular RL algorithms, including Group Relative Policy Optimization (GRPO) (Shao et al., 2024) and Proximal Policy Optimization (PPO) (Schulman et al., 2017), are applied in this setting. However, such single-turn formulation is inadequate for long-horizon multi-turn reasoning as it treats the entire trajectory as a single decision step, ignoring the multi-turn structure of the interactive tasks. In particular, it ignores intermediate signals that evaluate each complete agent-environment interaction, such as a tool call and its result, providing feedback at the granularity of a single turn in multi-turn tasks (Lightman et al., 2023; Zhang et al., 2025b; Ma et al., 2023; Choudhury, 2025). Without access to dense turn-level feedback, agents struggle to refine their behavior, making it difficult to interact effectively with dynamic environments over multiple

steps. For example, in a search agent (Chen et al., 2025b; Jin et al., 2025a), selecting a good query early on is crucial for retrieving relevant information; without turn-level retrieval feedback, the agent may not learn which queries contribute to correct answers.

Recent studies (Li et al., 2025a; Qian et al., 2025; Wang et al., 2025a; Labs, 2025; Wang et al., 2025b; Singh et al., 2025; Zhang et al., 2025a; Jin et al., 2025a) model multi-turn agentic tasks as Markov Decision Processes (MDPs) and incorporate intermediate rewards like tool execution. However, these approaches suffer from a credit assignment problem: they merge outcome and intermediate rewards into a sparse trajectory-level signal for RL training. This aggregation makes advantage estimation inaccurate and prevents RL algorithms from providing fine-grained supervision across intermediate rounds of interaction (Guo et al., 2025b; Feng et al., 2025b; Zhang et al., 2025c).

Motivated by this, we investigate turn-level reward design for both multi-turn RL algorithms and agent applications. Our key contributions are as follows:

- Most existing methods adopt MDP formulations with trajectory-level rewards, either terminal rewards that provide only a final outcome signal, or delayed rewards that merge intermediate and outcome signals into a single sparse feedback, leading to poor credit assignment. To address this limitation, we reformulate these tasks as MDPs with explicit turn-level rewards and provide theoretical analysis supporting the effectiveness of this design.

- To train multi-turn LLM agents effectively under our MDP formulation, we propose to extend GRPO and PPO to their multi-turn variants by incorporating both outcome and intermediate rewards, enabling fine-grained credit assignment. While multi-turn GRPO requires exponential rollout samples to compute intermediate advantages, multi-turn PPO leverages a critic model, offering a more efficient and scalable solution.

- To highlight the importance of turn-level rewards, we conduct a case study using a reasoning-augmented search agent that performs multiple rounds of reasoning and search before producing the final answer. We carefully design turn-level verifiable rewards and turn-level LLM-as-judge rewards for training the search agent. While verifiable rewards are rigid, the LLM-as-judge enables a more flexible and nuanced evaluation.

- Building on this case study, our experiments on multi-turn reasoning-augmented search tasks show that our proposed MDP formulation integrated turn-level rewards enables RL algorithms to significantly outperform baseline methods with trajectory-level rewards. Both training and validation reward curves obtained with the Qwen2.5-7B model demonstrate that our algorithm with turn-level rewards achieves more stable training, faster convergence, and higher accuracy for both verifiable and LLM-as-judge rewards. Furthermore, benchmarks on both in-domain and out-of-domain tasks show that our approach consistently achieves the highest accuracy and reliably produces outputs with 100% correct format.

## 2 PROBLEM FORMULATION FOR MULTI-TURN AGENT INTERACTION

### 2.1 TURN-LEVEL MDP FORMULATION

Let $x$ denote the input prompt sampled from the dataset $\mathcal{D}$, and $y = [l_1, f_1, \ldots, l_K, f_K]$ denote the complete output response, where $l_k$ is the response generated from an LLM policy $\pi_\theta$, and $f_k$ is the corresponding environment feedback at the $k$-th turn.

LLM agents operate in interactive environments where each turn yields stochastic feedback. To capture these dynamics, we formulate the multi-turn agentic task as a turn-level MDP, which is formally defined as $\mathcal{M} = \{\mathcal{S}, \mathcal{A}, P, R, \gamma\}$. Here, $\mathcal{S}$ denotes the state space, and $\mathcal{A}$ denotes the action space; A state $s \in \mathcal{S}$ typically corresponds to an interaction history, while an action $a \in \mathcal{A}$ often corresponds to a sequence of generated tokens; $P$ represents the transition dynamics; $R$ is the turn-level reward function; $\gamma$ is the discount factor. At the $k$-th turn, conditioned on the current state $s_k$, the agent makes an action $a_k$ according to the policy $\pi_\theta$, where $a_k = [l_k, f_k]$ if environment feedback exists, otherwise $a_k = l_k$. The agent then receives a turn-level reward $R_k = R(s_k, a_k)$, and transitions to the next state $s_{k+1}$. A multi-turn rollout trajectory is

$$\tau = \{(s_1, a_1, R_1), (s_2, a_2, R_2), \ldots, (s_K, a_K, R_K)\}$$

Note that the outcome reward is denoted by $R(x, y) = R_K$ for a prompt–response pair $(x, y)$.

## 2.2 Reward Assignment in Turn-Level MDP

Based on the granularity of reward assignment, we categorize multi-turn formulations into three types of turn-level MDPs:

1. Turn-level MDP with a terminal reward $\mathcal{M}_1$: provides only a final outcome (terminal) reward with no intermediate rewards.
2. Turn-level MDP with a delayed reward $\mathcal{M}_2$: provides an accumulated reward that merges both intermediate and outcome rewards into a single delayed signal.
3. Turn-level MDP with explicit turn-level rewards $\mathcal{M}_3$: provides explicit rewards at each turn.

Here, $\mathcal{M}_1$ contains only *outcome* rewards, whereas both $\mathcal{M}_2$ and $\mathcal{M}_3$ include *intermediate* rewards but differ in how these rewards are distributed across the turns. Moreover, $\mathcal{M}_1$ and $\mathcal{M}_2$ provide *trajectory-level* rewards, whereas $\mathcal{M}_3$ provides explicit *turn-level* rewards. Notably, most existing multi-turn agent studies adopt either $\mathcal{M}_1$ or $\mathcal{M}_2$. In contrast, our paper focuses on $\mathcal{M}_3$.

$\mathcal{M}_1$ is simple, and many existing studies adopt this formulation for multi-turn agentic tasks, relying on a final outcome reward such as answer correctness:

$$\max_{\pi_\theta} \; \mathbb{E}_{x\sim\mathcal{D},\, y\sim\pi_\theta(\cdot|x)} \left[ R(x,y) \right] \tag{1}$$

which can be interpreted as a contextual bandit problem (Bouneffouf & Feraud, 2025; Baheri & Alm, 2023). However, such a single-turn formulation is inadequate because it treats the entire trajectory as a single decision step and ignores intermediate rewards that capture the structure of agent–environment interactions. Without intermediate rewards, the system must simulate entire trajectories before receiving any feedback, leaving it unable to prune or down-weight clearly suboptimal trajectories at early stages (Wu et al., 2023; Singhal et al., 2025) and resulting in poor credit assignment.

$\mathcal{M}_2$ and $\mathcal{M}_3$ are return-equivalent, both maximizing the cumulative discounted return:

$$\max_{\pi_\theta} \; \mathbb{E}_{s_k,\, a_k\sim\pi_\theta(\cdot|s_k)} \left[ \sum_{k=1}^{K} \gamma^k R(s_k, a_k) \right] \tag{2}$$

in the sense that $\mathcal{M}_2$ and $\mathcal{M}_3$ have the same optimal $Q$-values and therefore share the same optimal policies. However, prior theoretical work (Ng et al., 1999; Arjona-Medina et al., 2019) indicates that when rewards are heavily delayed, $\mathcal{M}_2$ suffers from severe credit assignment issues, leading to high-variance gradients during policy optimization. See Appendix C for theoretical analysis.

# 3 GRPO with Turn-Level Rewards for Multi-Turn Agentic Tasks

## 3.1 Vanilla GRPO with Trajectory-Level Rewards

**GRPO.** Recently, the Group Relative Policy Optimization (GRPO) algorithm (Shao et al., 2024) has been widely used to enhance the reasoning capabilities of LLMs, which estimates the advantage in a group-relative manner. Specifically, for each input question $x$, it samples a group of responses $\{y_1, y_2, \ldots, y_G\}$ from the reference policy $\pi_{\text{ref}}$. GRPO optimizes the policy by maximizing the following objective function:

$$\mathcal{J}_{\text{GRPO}}(\theta) = \mathbb{E}_{x\sim\mathcal{D},\, \{y_i\}_{i=1}^{G}\sim\pi_{\text{old}}(\cdot|x)}$$

$$\left[ \frac{1}{G} \sum_{i=1}^{G} \frac{1}{|y_i|} \sum_{t=1}^{|y_i|} \min\left( w_{i,t}(\theta) A_{i,t}, \text{clip}\left( w_{i,t}(\theta), 1-\epsilon, 1+\epsilon \right) A_{i,t} \right) - \beta \mathbb{D}_{\text{KL}}\left[ \pi_\theta \,\|\, \pi_{\text{ref}} \right] \right], \tag{3}$$

where $w_{i,t}(\theta) = \frac{\pi_\theta(y_{i,t}|x,y_{i,<t})}{\pi_{\text{old}}(y_{i,t}|x,y_{i,<t})}$ is the token-level importance sampling ratio between the current policy $\pi_\theta$ and the previous policy $\pi_{\text{old}}$, $\epsilon$ is the clipping parameter, and $\beta$ is the KL divergence coefficient. Given a group of trajectory-level rewards $\{R_i^{\text{traj}}\}_{i=1}^{G}$, the advantage of the $i$-th response $A_{i,t}$ is calculated by

$$A_{i,t} = A_i^{\text{GRPO}} = \frac{R_i^{\text{traj}} - \text{mean}(\{R_i^{\text{traj}}\}_{i=1}^{G})}{\text{std}(\{R_i^{\text{traj}}\}_{i=1}^{G})} \tag{4}$$

GRPO is naturally suited for MDPs with trajectory-level rewards, namely $\mathcal{M}_1$ and $\mathcal{M}_2$, where $R_i^{\text{traj}} = R(x, y_i)$ for $\mathcal{M}_1$, and $R_i^{\text{traj}} = \sum_{k=1}^{K} \gamma^k R_{i,k}$ for $\mathcal{M}_2$, with $R_{i,k} = R(s_{i,k}, a_{i,k})$ denoting the intermediate reward given the state $s_{i,k}$ and action $a_{i,k}$ at the $k$-th turn.

**Limitations of GRPO in Multi-Turn Settings.** For GPRO, the advantage function $A_{i,t}$ in Eq. (4) is computed at the *trajectory level*, which means the same advantage is assigned uniformly across the entire trajectory, without distinguishing the contributions of individual turns or tokens. For long-horizon multi-turn tasks, such coarse-grained credit assignment often leads to unstable training and suboptimal performance (Guo et al., 2025b; Feng et al., 2025b; Zhang et al., 2025c).

### 3.2 MT-GPRO: Turn-Level Credit Assignment for GRPO

**MT-GPRO.** To highlight the importance of fine-grained credit assignment for GRPO, we consider a simple two-turn agent setting ($K = 2$), where the agent receives a group of intermediate rewards $\{R_i^I\}_{i=1}^G$ in the first turn and outcome rewards $\{R_i^O\}_{i=1}^G$ in the second turn. Based on these signals, we present our turn-level credit assignment strategy for GRPO. The resulting turn-level advantages in the first and second turns are given by:

$$A_{i,1}^{\text{MT-GPRO}} = A_i^I + \alpha A_i^O, \quad A_{i,2}^{\text{MT-GPRO}} = A_i^O, \tag{5}$$

where $A_i^I$ and $A_i^O$ denote the intermediate and outcome advantages:

$$A_i^I = \frac{R_i^I - \text{mean}(\{R_i^I\}_{i=1}^G)}{\text{std}(\{R_i^I\}_{i=1}^G)}, \quad A_i^O = \frac{R_i^O - \text{mean}(\{R_i^O\}_{i=1}^G)}{\text{std}(\{R_i^O\}_{i=1}^G)} \tag{6}$$

By leveraging intermediate rewards, all tokens within a single turn share a unified advantage signal. Moreover, the advantage of a turn depends not only on the rewards from that turn but also on the contributions of subsequent turns. We refer to this algorithm as *multi-turn GRPO (MT-GRPO)*. A detailed derivation of MT-GRPO for the $K$-turn setting is provided in Appendix F.

**Case Study for MT-GRPO on a Two-Turn Agentic Task.** We conduct experiments to evaluate the proposed MT-GRPO method in a two-turn agent setting, where the agent first calls the search tool with reasoning in the initial turn and then produces the final answer in the subsequent turn (see Appendix G for details). Beyond the outcome-level exact-match reward, we design intermediate rewards based on tool-execution feedback for MT-GRPO. Figure 1 presents training reward curves for GRPO and MT-GRPO, which show that MT-GRPO achieves more stable tool usage (left figure) and higher exact-match accuracy (right figure), highlighting the importance of fine-grained credit assignment in multi-turn agentic tasks.

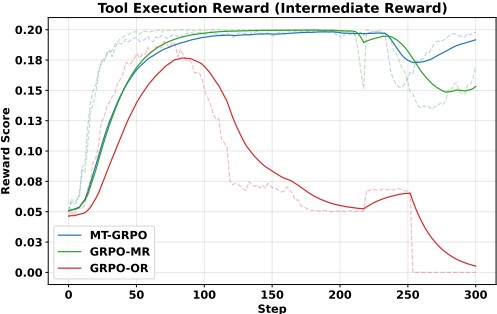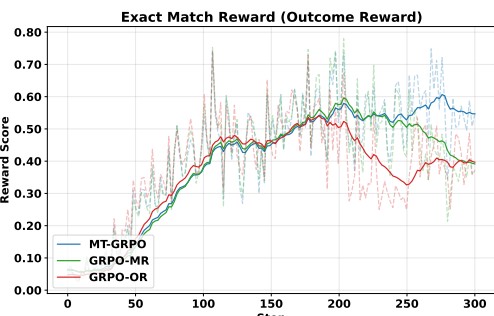

Figure 1: Curves for different training reward components during training with various algorithms. GRPO-OR means GRPO with outcome rewards while GPRO-MR means GRPO with merged outcome and intermediate rewards. GPRO-OR, GPRO-MR, and MT-GPRO correspond to the MDPs $\mathcal{M}_1$, $\mathcal{M}_2$, and $\mathcal{M}_3$, respectively. Each plot shows the training reward score over training steps. Dotted lines represent the average reward across 10 runs, while solid lines show trends smoothed using the Exponential Moving Average (EMA).

*Remark.* MT-GPRO has two limitations: (1) In MT-GRPO, computing the intermediate advantages requires $G$ rollout samples at each turn. Therefore, over a horizon of $K$ turns, this results in $G^{K-1}$ rollout trajectories in total (see Appendix F for details). Such *exponential* growth in complexity makes

the approach computationally prohibitive for long-horizon multi-turn tasks. (2) This strategy also assumes that all rollout samples in a group must contain *the same number of turns*, which requires enforcing this constraint in the system prompt and leads to a fixed-turn setting. Such a restriction limits the flexibility and applicability of GRPO in more diverse scenarios. For example, in a search task, one question may be resolved in a single tool call or require multiple calls to retrieve, filter, and refine results in a sampled group.

## 4 PPO WITH TURN-LEVEL REWARDS FOR MULTI-TURN AGENTIC TASKS

In the previous section, we illustrated the importance of fine-grained credit assignment for GPRO, which improves the performance of LLM agents in multi-turn interactions. However, the exponential computational cost, together with the fixed-turn constraint, limits the applicability of MT-GRPO to general agentic tasks. In this section, we present the PPO alogrithm with turn-level rewards, aiming to provide a more flexible, scalable, and efficient solution.

**PPO.** Proximal Policy Optimization (PPO) (Schulman et al., 2017) is a popular actor-critic RL algorithm commonly used for LLM training (Ouyang et al., 2022). PPO updates the policy by maximizing the following surrogate objective:

$$\mathcal{J}_{\text{PPO}}(\theta) = \mathbb{E}_{x \sim \mathcal{D}, \, y \sim \pi_{\text{old}}(\cdot|x)} \left[ \frac{1}{|y|} \sum_{t=1}^{|y|} \min \left( w_t(\theta) A_t, \, \text{clip} \left( w_t(\theta), \, 1 - \epsilon, \, 1 + \epsilon \right) A_t \right) \right], \quad (7)$$

The advantage estimate $A_t$ is computed using Generalized Advantage Estimation (GAE) (Schulman et al., 2015), based on rewards and a learned value function (critic model). Formally, for a trajectory of length $T$, the GAE $A_t$ at time step $t$ is computed as:

$$A_t = \sum_{l=0}^{T-t-1} (\gamma \lambda)^l \delta_{t+l}, \quad \delta_t = r_t + \gamma V_{t+1} - V_t \quad (8)$$

where $\gamma$ is the discount factor, $\lambda \in [0, 1]$ is the GAE parameter, $\delta_t$ is the temporal-difference error, $r_t$ is the token-level reward and $V_t$ is the token-level value at step $t$. Through the mechanism of GAE, the token-level value function enables token-level advantage estimation.

**Turn-Level Reward Assignment for PPO.** With explicit intermediate rewards, GAE provides fine-grained training signals at each turn. Given both intermediate rewards $R^I$ and the outcome reward $R^O$, the token-level reward $r_t$ is assigned as

$$r_t = \begin{cases} R^O & \text{if } t \text{ is the last token of the entire trajectory} \\ R^I & \text{if } t \text{ is the last token of the intermediate turn} \\ 0 & \text{otherwise} \end{cases} \quad (9)$$

For clarity, we refer to PPO trained with both intermediate and outcome rewards as *multi-turn PPO (MT-PPO)*, while PPO trained with only a sparse trajectory-level reward is referred to as *PPO*. To achieve fine-grained credit assignment with the usage of turn-level rewards, compared to MT-GRPO, which requires exponential rollout samples to compute intermediate advantages, MT-PPO leverages a critic model with GAE, offering a more efficient and scalable solution.

**Summary.** Table 1 summarizes the granularity of reward assignment and advantage estimation across different RL algorithms. As shown, MT-PPO provides fine-grained turn-level rewards and token-level advantage estimation. This higher granularity enables more precise credit assignment, which is particularly beneficial for multi-turn LLM agents where successful outcomes often depend on a sequence of intermediate decisions. In contrast, trajectory-level methods provide coarser feedback, which often leads to weaker learning signals and unstable training. These insights will be empirically validated in the following experiments.

Table 1: Comparison of granularity of reward assignment and advantage estimation across different RL algorithms for multi-turn LLM agents.

| RL Algo. | Granularity | |
|---|---|---|
| | Reward | Advantage |
| GRPO | Trajectory-Level | Trajectory-Level |
| MT-GRPO | Turn-Level | Turn-Level |
| PPO | Trajectory-Level | Token-Level |
| MT-PPO | Turn-Level | Token-Level |

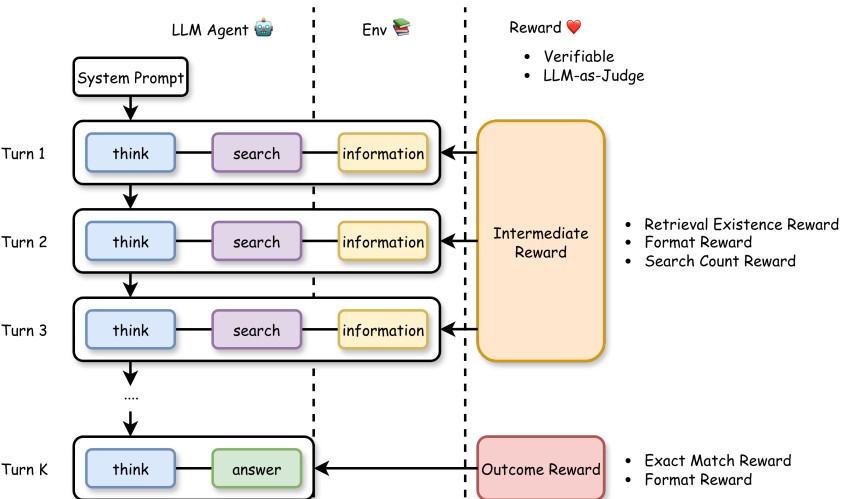

Figure 2: Overview of the multi-turn reasoning-augmented search agent pipeline. Given a system prompt and a question, each iteration of the LLM-based search agent proceeds as follows: (1) The agent begins with *reasoning*, analyzing the current context to identify missing information. (2) It then formulates a search query to *retrieve* relevant information from an external database, which is integrated into the evolving *context*. (3) This cycle continues until the agent judges that the context is sufficient, at which point it performs a final round of *reasoning* to generate the answer.

## 5 CASE STUDY: MULTI-TURN REASONING-AUGMENTED SEARCH AGENT

### 5.1 TASK FORMULATION

We study an LLM agent that performs multi-turn reasoning with search engine interactions. The task can be naturally formulated under the turn-level MDP framework, as discussed in Section 2, where each action $a$ corresponds either to a reasoning-augmented search step or to producing the final answer. The goal is to improve the agent's performance through effective integration of external search. Specifically, the agent learns to leverage a Wikipedia search engine to retrieve relevant information and generate an accurate answer. Without search calling, the agent must rely solely on its internal knowledge to answer questions, which can limit accuracy, especially for fact-based queries requiring up-to-date or domain-specific information. The overall interaction follows a multi-turn reasoning–search loop, as shown in Figure 2.

### 5.2 TURN-LEVEL VERIFIABLE REWARD DESIGN

Unlike existing approaches (Chen et al., 2025b; Jin et al., 2025b), which assign binary rewards based solely on final-answer correctness for the entire trajectory, we design turn-level verifiable rewards for both intermediate and final turns to better align with the environment of the LLM-based search agent.

**Outcome Verifiable Rewards** evaluate the model-generated responses in the last turn, focusing on both the correctness of the answer and the adherence to the required output format.

- *Outcome Exact Match Reward* evaluates whether the extracted answer (from the `<answer>` tag) exactly matches any accepted ground-truth answer after normalization (e.g., lowercasing and whitespace removal):

- *Outcome Format Reward* ensures format correctness by verifying that: (1) only `<think>` and `<answer>` tags appear (no extra tags), (2) each tag appears exactly once, and (3) `<think>` precedes `<answer>`.

The outcome reward is defined as

$$R^O = \begin{cases} 1 & f_{\text{em}} = \text{True}, \ f_{\text{format}} = \text{True}, \\ 0.2 & f_{\text{em}} = \text{False}, \ f_{\text{format}} = \text{True}, \\ -1 & f_{\text{format}} = \text{False}, \end{cases}$$

where $f_{\text{em}}$ and $f_{\text{format}}$ denote the indicators of answer (exact-match) correctness and format correctness, respectively. A smaller positive reward is given when the answer is incorrect but the output follows the required format, encouraging structural correctness during training. A negative reward (penalty) is applied when the format is incorrect, ensuring that the agent adheres to the required structure.

**Intermediate Verifiable Rewards** guide the agent's behavior in intermediate turns by evaluating the presence of ground-truth answers in retrieved content, enforcing proper format usage, and discouraging excessive search calls.

- *Intermediate Retrieval Existence Reward* evaluates whether any accepted answer appears in the one-round search result (from `<information>` tag), using case-insensitive matching. $R^I_{\text{retrieval}} = 0.3$ if retrieved information contains any ground-truth, otherwise 0.
- *Intermediate Format Reward* ensures format correctness by verifying that: (1) only `<think>`, `<search>`, and `<information>` tags appear (no extra tags), (2) each tag appears exactly once, and (3) `<think>` precedes `<search>` and `<information>`. $R^I_{\text{format}} = 0.1$ if the format is correct, otherwise $-0.2$.
- *Intermediate Search Count Reward* penalizes excessive search usage.

$$R^I_{\text{search}} = -\lambda_s \cdot n_{\text{search}},$$

where $\lambda_s$ is a predefined positive constant controlling the weight of the search count reward, $n_{\text{search}}$ denotes the cumulative number of search invocations from the first turn up to the current turn.

The intermediate reward is defined as $R^I = R^I_{\text{retrieval}} + R^I_{\text{format}} + R^I_{\text{search}}$. Retrieval correctness is similarly assigned a smaller weight than answer correctness, again to reduce the risk of reward hacking. In addition, we introduce an intermediate search penalty to discourage excessive or unnecessary search calls, preventing the agent from either avoiding the avoiding the question answering or failing due to crashes.

## 5.3 LLM AS JUDGE FOR TURN-LEVEL EVALUATION

Verifiable rewards, such as exact match, provide a strict and objective form of evaluation. However, they can be overly rigid: an agent may produce a correct answer that differs slightly in form from the ground truth but still receives negative feedback. To complement such verifiable signals, we adopt the *LLM-as-judge* paradigm, where a strong LLM evaluates agent outputs. The LLM-as-judge framework consists of two key components: step-by-step reasoning and rubric-based scoring.

**Reasoning.** We employ a generative reasoning model (GRM) (Li et al., 2025b) as the judge, prompting it to generate detailed justifications before assigning a score. The step-by-step reasoning process encourages the judge to evaluate output quality using rubric-based criteria rather than relying on shallow correlations.

**Rubrics.** Rubric-based scoring provides structured evaluation criteria that improve both consistency and reliability across assessments. Unlike outcome-level evaluation that only considers the final answer, our framework assesses each turn's output. This fine-grained assessment offers richer feedback and aligns naturally with multi-turn agentic tasks, where intermediate steps critically influence overall success. The judge model evaluates format correctness, reasoning quality, and search effectiveness, while also applying a search penalty to discourage excessive or unnecessary tool calls. Additional implementation details are provided in Appendix D.2.

## 6 EXPERIMENTS

In our experiments, we build our codebase upon the open-source project Search-R1 (Jin et al., 2025b), which trains LLM agents for multi-turn reasoning-augmented search tasks. More details on experimental settings can be found in Appendix D.1.

## 6.1 EVALUATED METHODS

We compare both training reward dynamics and benchmark performance across different methods.

**Training Dynamics.** We evaluate our MT-PPO against several PPO-based baselines:

- PPO-OR (Jin et al., 2025b): vanilla PPO trained with only outcome rewards, where the trajectory-level reward is a binary signal indicating final-answer correctness, corresponding to the terminal-reward MDP $\mathcal{M}_1$.
- PPO-MR (Jin et al., 2025a): vanilla PPO trained with merged intermediate and outcome rewards, where the trajectory-level reward combines intermediate rewards (retrieval correctness) and outcome rewards (answer correctness and format correctness), corresponding to the delayed-reward MDP $\mathcal{M}_2$. The detailed reward design is provided in Section 4.1 of (Jin et al., 2025a).
- MT-PPO (ours): PPO variant trained with both intermediate and outcome rewards, where the turn-level reward design is described in Section 5.2, with $\lambda_s = 0.1$ by default, corresponding to the turn-level-reward MDP $\mathcal{M}_3$.

We omit GRPO training curves since, as reported in (Jin et al., 2025b), GRPO consistently crashes during training.

**Benchmark Evaluation.** In addition to the base model and the instruct model, we further compare our method against Search-R1 trained with GRPO and PPO (Jin et al., 2025b;a),[1] OTC trained with GRPO and PPO (Wang et al., 2025a), and StepSearch trained with PPO (Wang et al., 2025c).

Since PPO baselines often crash, we evaluate them using either the final checkpoint or the last checkpoint prior to collapse.

**Evaluation Metrics.** We evaluate model performance using three types of rewards: (1) answer correctness (exact match) reward, (2) format correctness reward, and (3) retrieval correctness reward. Each reward is assigned a value of 1.0 if the criterion is satisfied and 0 otherwise.

## 6.2 EXPERIMENT SETUP

**Datasets.** These datasets are categorized as follows: (1) General Question Answering: NQ (Karpukhin et al., 2020), TriviaQA (Joshi et al., 2017), and PopQA (Mallen et al., 2022). (2) Multi-Hop Question Answering: HotpotQA (Yang et al., 2018), 2WikiMultiHopQA (Ho et al., 2020), and Musique (Trivedi et al., 2022). These datasets cover a diverse range of search and reasoning challenges, providing a comprehensive basis for evaluation.

**Training Details.** We use Qwen2.5-7B (Yang et al., 2024) as the base model, E5 (Wang et al., 2022) as the retriever, and 2018 Wikipedia dump (Karpukhin et al., 2020) as the corpus. We set the number of retrieved passages to 3, and the maximum number of turns $N_{\max}$ to 4. The system prompt follows that of Search-R1 (Jin et al., 2025b). We also enable policy loss masking on retrieved tokens.

## 6.3 MAIN RESULTS

**Training Dynamics.** Figures 3 and 5 show training and validation reward curves for PPO and MT-PPO. MT-PPO achieves substantially more stable training, converging faster in the early phase (first 100 steps) thanks to intermediate rewards that provide stronger guidance. As training progresses, PPO exhibits high variance and even performance degradation, especially on HotpotQA, while MT-PPO maintains consistent improvement. MT-PPO attains higher average accuracy than PPO, demonstrating greater robustness. Format reward curves show that MT-PPO consistently follows the correct output format, while PPO struggles, especially on HotpotQA, where formatting mistakes prevent correct evaluation. This indicates that turn-level rewards in MT-PPO stabilize training and enforce structural correctness. Retrieval curves further show that MT-PPO achieves more consistent accuracy by leveraging intermediate signals to guide reasoning. Figure 6 presents training curves for MT-PPO and PPO with judge rewards, where MT-PPO again demonstrates stable optimization.

---

[1]The GRPO baselines (GRPO-OR and GRPO-MR) correspond to the PPO baselines (PPO-OR and PPO-MR) with the same reward design (Jin et al., 2025b;a).

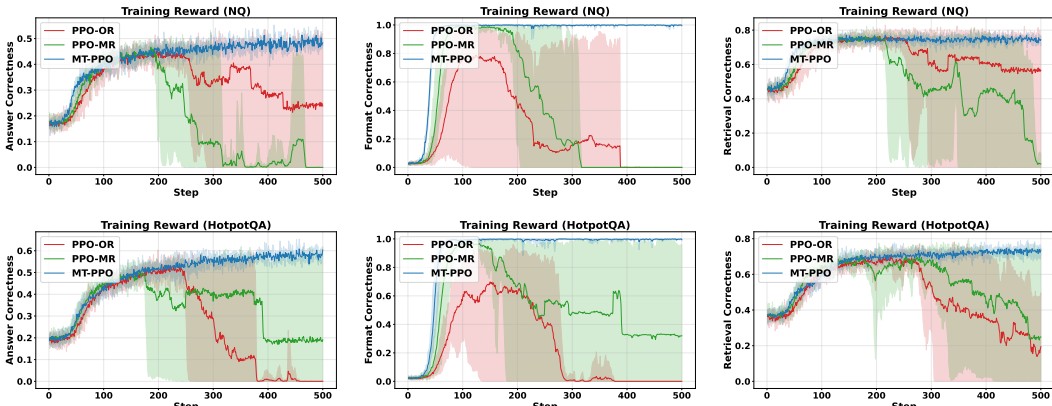

Figure 3: Training reward curves recorded during training for PPO baselines and MT-PPO on the NQ and HotpotQA datasets. The rewards include answer correctness, format correctness, and retrieval correctness. Solid lines show mean reward values, while shaded regions indicate variability across five independent runs.

Table 2: The performance results of different methods on six datasets. Bold numbers indicate the best performance for each dataset and metric. [†]/[*] denote in-domain/out-of-domain datasets. [‡] indicates results copied from the original paper.

| Methods | General QA | | | Multi-Hop QA | | | Avg. |
|---|---|---|---|---|---|---|---|
| | NQ[†] | TriviaQA* | PopQA* | HotpotQA[†] | 2wiki* | Musique* | |
| *Answer Correctness (Exact Match)* | | | | | | | |
| Qwen2.5-7B-Base | 0.177 | 0.319 | 0.181 | 0.160 | 0.167 | 0.040 | 0.174 |
| Qwen2.5-7B-Instruct | 0.320 | 0.563 | 0.349 | 0.292 | 0.277 | 0.118 | 0.320 |
| GRPO-OR (Search-R1) | 0.391 | 0.560 | 0.388 | 0.331 | 0.306 | 0.129 | 0.351 |
| GRPO-MR (Search-R1)[‡] | 0.453 | 0.628 | 0.450 | 0.416 | 0.375 | 0.164 | 0.414 |
| PPO-OR (Search-R1) | 0.483 | 0.639 | 0.456 | 0.435 | 0.382 | 0.199 | 0.432 |
| PPO-MR (Search-R1)[‡] | 0.472 | 0.629 | 0.452 | 0.436 | 0.402 | 0.180 | 0.429 |
| GRPO (OTC)[‡] | 0.444 | 0.597 | 0.431 | 0.366 | 0.311 | 0.130 | 0.380 |
| PPO (OTC)[‡] | 0.446 | 0.623 | 0.425 | 0.383 | 0.363 | 0.152 | 0.399 |
| PPO (StepSearch) | 0.355 | 0.570 | 0.385 | 0.351 | 0.396 | 0.179 | 0.373 |
| MT-PPO (ours) | **0.490** | **0.647** | **0.459** | **0.453** | **0.424** | **0.209** | **0.447** |
| *Format Correctness* | | | | | | | |
| Qwen2.5-7B-Base | 0.118 | 0.118 | 0.105 | 0.098 | 0.084 | 0.082 | 0.101 |
| Qwen2.5-7B-Instruct | 0.183 | 0.267 | 0.067 | 0.109 | 0.037 | 0.071 | 0.122 |
| GRPO-OR (Search-R1) | 0.706 | 0.685 | 0.597 | 0.513 | 0.376 | 0.328 | 0.534 |
| PPO-OR (Search-R1) | 0.909 | 0.954 | 0.952 | 0.916 | 0.806 | 0.834 | 0.895 |
| PPO (StepSearch) | 0.521 | 0.614 | 0.668 | 0.560 | 0.396 | 0.571 | 0.555 |
| MT-PPO (ours) | **0.999** | **0.997** | **0.999** | **0.998** | **0.999** | **0.999** | **0.999** |

**Benchmark Performance.** Table 2 reports results on six QA datasets, spanning both general and multi-hop reasoning tasks. MT-PPO consistently outperforms PPO and GRPO in answer correctness, with the largest gains on multi-hop tasks such as HotpotQA and 2Wiki. Moreover, MT-PPO nearly perfects format correctness, reaching close to 100% across datasets, underscoring the effectiveness of multi-turn credit assignment in producing both accurate and well-structured outputs.

## 6.4 ABLATION STUDY

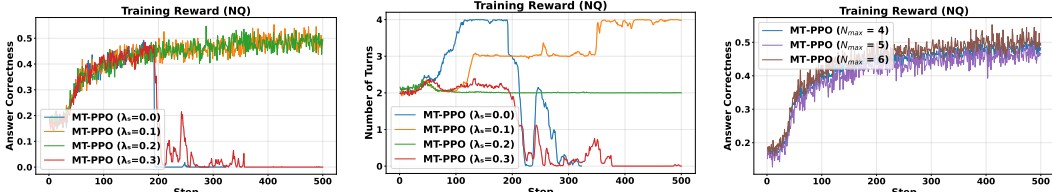

Figure 4: Ablation studies on (1) the search count reward $\lambda_s$ and (2) the maximum number of turns $N_{\max}$ on the NQ dataset. The left panel reports answer correctness, the middle panel shows the average number of turns, and the right panel illustrates accuracy under different $N_{\max}$ settings.

We conduct two ablation studies to analyze the effects of (1) the search-count reward and (2) the maximum number of turns on training dynamics and final performance. As shown in Figure 4, incorporating a moderate search-count reward (e.g., $\lambda_s = 0.1$) significantly improves training stability and answer correctness. The left panel shows that MT-PPO with $\lambda_s = 0.1$ achieves the highest and most consistent accuracy, while overly strong penalties (e.g., $\lambda_s = 0.3$) degrade performance.

The middle panel illustrates how the search-count reward shapes the agent's turn usage. With $\lambda_s = 0.1$, the agent learns to reduce unnecessary search calls early in training and eventually stabilizes around an efficient number of turns. In contrast, removing this term ($\lambda_s = 0.0$) leads to unstable behavior, including excessive or erratic turn usage, which ultimately harms convergence.

Finally, the right panel shows the effect of varying the maximum number of allowed turns $N_{\max}$. The results indicate that MT-PPO is robust across different turn limits: adjusting $N_{\max}$ from 4 to 6 yields nearly identical accuracy curves. This suggests that MT-PPO adapts its strategy effectively without being overly sensitive to the chosen turn budget.

## 7 CONCLUSION AND FUTURE WORK

In this paper, we highlighted the importance of turn-level rewards for multi-turn agentic tasks. By introducing carefully designed intermediate signals, we extended GRPO and PPO into multi-turn variants, allowing LLM agents to receive more informative feedback at each stage of interaction. Experiments on reasoning-augmented search agents show that incorporating turn-level rewards substantially improves both the stability and accuracy of training across different RL algorithms. We believe that turn-level rewards have broad applicability beyond search, offering a general mechanism for improving the effectiveness of multi-turn agents in diverse interactive environments.

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

## A    LLM Usage

In this work, LLMs were used exclusively for polishing the writing. No part of the technical content, experimental design, or analysis relied on LLMs. The authors retain full responsibility for the correctness and originality of the ideas, methods, and results.

## B    Related Work

### B.1    Process Reward and Credit Assignment in RL

Process rewards provide fine-grained credit assignment and enhance both training efficiency and optimization stability in RL. Such dense rewards have proven effective in classical RL domains such as games and robotic control (Schrittwieser et al., 2020; Liu et al., 2022; Sun et al., 2025). Process reward models have also been extensively explored for inference-time scaling in LLMs (Lightman et al., 2023; Uesato et al., 2022). Recent studies have further highlighted the importance of effective credit assignment in RL (Pignatelli et al., 2023; Shao et al., 2024; Cui et al., 2025; Cheng et al., 2025; Feng et al., 2025b; Guo et al., 2025b), particularly for textual reasoning tasks such as mathematical problem solving. In multi-turn agent interaction settings, turn-level rewards evaluate each complete agent–environment interaction, such as a tool invocation and its resulting output, providing feedback at the granularity of a single turn. This setting naturally emphasizes the advantages of process-level rewards and fine-grained credit assignment. However, the design of effective reward functions for multi-turn agents, as well as RL algorithms capable of leveraging such fine-grained credit signals, remains underexplored. A recent study (Chen et al., 2025c) investigates step-level credit assignment for multi-turn LLM agents by introducing a dual-discounting GAE formulation in PPO. While their method relies on critic-derived step-level value estimates for credit assignment, our approach directly incorporates explicit intermediate rewards into the PPO objective, enabling more precise and more stable credit assignment.

### B.2    RL for LLM Agents

RL has been applied to train long-horizon multi-turn LLM agents in diverse domains, including search (Chen et al., 2025b; Jin et al., 2025b;a), tool use (Feng et al., 2025a; Li et al., 2025a; Qian et al., 2025; Wang et al., 2025a; Labs, 2025; Zhang et al., 2025a; Singh et al., 2025), text-based games (Yao et al., 2020; Carta et al., 2023; Zhai et al., 2024; Wang et al., 2025b), web shopping (Yao et al., 2022), digital app interaction (Chen et al., 2025a), and mobile device control (Bai et al., 2024). A number of these studies (Jin et al., 2025a; Feng et al., 2025a; Li et al., 2025a; Qian et al., 2025; Wang et al., 2025a; Labs, 2025; Zhang et al., 2025a; Singh et al., 2025) apply RL algorithms such as GRPO and PPO to train tool-using LLM agents, including calculators, code interpreters, and search engines, thus enabling reasoning with external tools. However, these methods generally collapse outcome- and turn-level signals into a single trajectory-level reward, limiting fine-grained credit assignment. One related work is StepSearch (Wang et al., 2025c), which applies PPO with turn-level rewards for multi-turn search. However, it relies heavily on data augmentation and requires prompt modifications during preprocessing. In contrast, our method avoids such prompt engineering and provides a cleaner, more general framework for turn-level reward design. Our approach is orthogonal to existing search-agent methods (Chen et al., 2025b; Jin et al., 2025b;a; Wang et al., 2025c) and applies broadly to multi-turn LLM agents beyond search tasks.

## C    Theoretical Analysis of Policy-Gradient Variance

In this section, we analyze and compare the policy-gradient variance of the two return-equivalent MDPs $\mathcal{M}_2$ and $\mathcal{M}_3$, since the comparison between $\mathcal{M}_1$ and $\mathcal{M}_3$ is trivial. Although $\mathcal{M}_2$ and $\mathcal{M}_3$ share the same optimal $Q$-values and thus the same optimal policies, they differ in how rewards are assigned across turns. Our theoretical results show that the MDP with explicit turn-level rewards $\mathcal{M}_3$ yields lower policy-gradient variance than the MDP that relies solely on a delayed accumulated reward $\mathcal{M}_2$. These findings help justify our formulation and highlight why turn-level rewards and fine-credit credit assignment leads to more stable and efficient RL training for GRPO and PPO, as demonstrated in Sections 3 and 4.

In general, the policy gradient of the expected return objective $J(\theta)$ can be written as

$$\nabla_\theta J(\theta) = \mathbb{E}_{\tau \sim \pi_\theta} \left[ \sum_{k=1}^{K} \nabla_\theta \log \pi_\theta(a_k \mid s_k) \, G_k \right],$$

where $G_k$ denotes the return associated with timestep $k$. For the delayed-reward MDP $\mathcal{M}_2$, the timestep return is

$$G_k = \sum_{t=1}^{K} \gamma^t R(s_t, a_t),$$

which is the discounted sum of all turn-level rewards over the trajectory. For the turn-level-reward MDP $\mathcal{M}_3$, the timestep return is

$$G_k = \sum_{t=k}^{K} \gamma^{t-k} R(s_t, a_t),$$

which corresponds to the discounted future return starting from timestep $k$.

**Lemma 1.** *Consider a $K$-step episodic MDP with random rewards $R_1, \ldots, R_K$. Let*

$$h_k = \nabla_\theta \log \pi_\theta(a_k \mid s_k), \qquad k = 1, \ldots, K.$$

*Suppose that the discount factor is $\gamma = 1$, and define the returns*

$$G_k^{\mathcal{M}_2} = \sum_{t=1}^{K} R_t, \qquad G_k^{\mathcal{M}_3} = \sum_{t=k}^{K} R_t.$$

*The REINFORCE estimator is*

$$\hat{g} = \sum_{k=1}^{K} h_k G_k,$$

*where $G_k$ can be chosen as either $G_k^{\mathcal{M}_2}$ or $G_k^{\mathcal{M}_3}$. Assume:*

*(i) the rewards $(R_1, \ldots, R_K)$ are independent of $(h_1, \ldots, h_K)$;*

*(ii) each $h_k$ has finite mean and variance, with $\mathbb{E}[h_k] = m_k$ and $\mathrm{Var}(h_k) < \infty$;*

*(iii) $\mathrm{Cov}(h_i G_i, h_j G_j) = 0$ for all $i \neq j$;*

*(iv) the rewards are nonnegative and positively correlated;*

*(v) each reward has finite mean and variance,*

$$\mathbb{E}[R_t] = \mu_t, \qquad \mathrm{Var}(R_t) = \sigma_t^2.$$

*Then the following holds:*

$$\mathrm{Var}\big(\hat{g}^{\mathcal{M}_2}\big) \ \geq \ \mathrm{Var}\big(\hat{g}^{\mathcal{M}_3}\big),$$

*and the variance gap admits the explicit lower bound*

$$\mathrm{Var}\big(\hat{g}^{\mathcal{M}_2}\big) - \mathrm{Var}\big(\hat{g}^{\mathcal{M}_3}\big) \ \geq \ \sum_{k=1}^{K} \left[ \mathrm{Var}(h_k)\Big(\sum_{t=1}^{k-1} \sigma_t^2 + \big(\sum_{t=1}^{k-1} \mu_t\big)^2\Big) + m_k^2 \sum_{t=1}^{k-1} \sigma_t^2 \right] \ \geq \ 0.$$

*Proof.* We begin by computing the variance of $h_k G_k$:

$$\mathrm{Var}(h_k G_k) = \mathbb{E}[h_k^2 G_k^2] - \big(\mathbb{E}[h_k G_k]\big)^2.$$

Because $h_k$ and $G_k$ are independent by Assumption (i),

$$\mathbb{E}[h_k^2 G_k^2] = \mathbb{E}[h_k^2]\,\mathbb{E}[G_k^2], \qquad \mathbb{E}[h_k G_k] = \mathbb{E}[h_k]\,\mathbb{E}[G_k] = m_k\,\mathbb{E}[G_k].$$

Using $\mathbb{E}[h_k^2] = \mathrm{Var}(h_k) + m_k^2$ gives

$$\begin{aligned}
\mathrm{Var}(h_k G_k) &= \mathbb{E}[h_k^2] \, \mathbb{E}[G_k^2] - m_k^2 \, \mathbb{E}[G_k]^2 \\
&= \left(\mathrm{Var}(h_k) + m_k^2\right) \mathbb{E}[G_k^2] - m_k^2 \, \mathbb{E}[G_k]^2 \\
&= \mathrm{Var}(h_k) \, \mathbb{E}[G_k^2] + m_k^2 \, \mathbb{E}[G_k^2] - m_k^2 \, \mathbb{E}[G_k]^2 \\
&= \mathrm{Var}(h_k) \, \mathbb{E}[G_k^2] + m_k^2 \left(\mathbb{E}[G_k^2] - \mathbb{E}[G_k]^2\right) \\
&= \mathrm{Var}(h_k) \, \mathbb{E}[G_k^2] + m_k^2 \, \mathrm{Var}(G_k),
\end{aligned}$$

For $\mathcal{M}_2$ and $\mathcal{M}_3$, we have

$$G_k^{\mathcal{M}_2} = S = \sum_{t=1}^{K} R_t, \qquad G_k^{\mathcal{M}_3} = S_k = \sum_{t=k}^{K} R_t.$$

Thus

$$\mathrm{Var}(h_k G_k^{\mathcal{M}_2}) = \mathrm{Var}(h_k) \, \mathbb{E}[S^2] + m_k^2 \, \mathrm{Var}(S),$$
$$\mathrm{Var}(h_k G_k^{\mathcal{M}_3}) = \mathrm{Var}(h_k) \, \mathbb{E}[S_k^2] + m_k^2 \, \mathrm{Var}(S_k).$$

Subtracting yields

$$\mathrm{Var}(h_k G_k^{\mathcal{M}_2}) - \mathrm{Var}(h_k G_k^{\mathcal{M}_3}) = \mathrm{Var}(h_k)\left(\mathbb{E}[S^2] - \mathbb{E}[S_k^2]\right) + m_k^2\left(\mathrm{Var}(S) - \mathrm{Var}(S_k)\right). \quad (\star)$$

Next we characterize the terms in $(\star)$. Define the prefix sum

$$P_k = \sum_{t=1}^{k-1} R_t, \qquad S = P_k + S_k.$$

A direct expansion gives

$$\mathbb{E}[S^2] - \mathbb{E}[S_k^2] = \mathbb{E}[P_k^2] + 2\mathbb{E}[P_k S_k].$$

Since rewards are nonnegative and positively correlated by Assumption (iv),

$$\mathbb{E}[P_k S_k] \geq 0.$$

Furthermore, Assumption (v) implies

$$\mathbb{E}[P_k^2] = \mathrm{Var}(P_k) + (\mathbb{E}[P_k])^2 \geq \sum_{t=1}^{k-1} \sigma_t^2 + \left(\sum_{t=1}^{k-1} \mu_t\right)^2.$$

Similarly,

$$\mathrm{Var}(S) - \mathrm{Var}(S_k) = \mathrm{Var}(P_k) + 2\mathrm{Cov}(P_k, S_k) \geq \mathrm{Var}(P_k) \geq \sum_{t=1}^{k-1} \sigma_t^2.$$

Substituting these lower bounds into $(\star)$ yields

$$\mathrm{Var}(h_k G_k^{\mathcal{M}_2}) - \mathrm{Var}(h_k G_k^{\mathcal{M}_3}) \geq \mathrm{Var}(h_k) \left(\sum_{t=1}^{k-1} \sigma_t^2 + \left(\sum_{t=1}^{k-1} \mu_t\right)^2\right) + m_k^2 \sum_{t=1}^{k-1} \sigma_t^2.$$

Because $\mathrm{Cov}(h_i G_i, h_j G_j) = 0$ for $i \neq j$ by Assumption (iii), the variance of each estimator decomposes into the sum of its per-step variances:

$$\mathrm{Var}\left(\hat{g}^{\mathcal{M}_2}\right) = \sum_{k=1}^{K} \mathrm{Var}\left(h_k G_k^{\mathcal{M}_2}\right), \qquad \mathrm{Var}\left(\hat{g}^{\mathcal{M}_3}\right) = \sum_{k=1}^{K} \mathrm{Var}\left(h_k G_k^{\mathcal{M}_3}\right).$$

Since we have shown that each term satisfies

$$\mathrm{Var}\left(h_k G_k^{\mathcal{M}_2}\right) \geq \mathrm{Var}\left(h_k G_k^{\mathcal{M}_3}\right),$$

with an explicit lower bound on the difference, summing over $k = 1, \ldots, K$ immediately yields both the overall variance ordering and the stated lower bound. This concludes the proof. $\qquad \square$

The assumptions used in Lemma 1 are mild. Assumptions (ii) and (v) impose finite first and second moments on $h_k$ and $R_k$, which are required to ensure that all variance terms are well defined and to derive the explicit variance gap. Assumption (i) reflects the natural fact that rewards are generated by the environment and do not directly depend on the stochasticity of the policy gradient estimator. Assumption (iii) holds when different time steps use independent sampling noise and allows the variance of the full estimator to decompose into a sum of per-step variances. Finally, Assumption (iv) states that rewards are nonnegative and positively correlated, a property satisfied in many episodic tasks where progress or success accumulates over time.

## D  PPO EXPERIMENTS

### D.1  DETAILS FOR EXPERIMENTAL SETUP (PPO)

#### D.1.1  EVALUATED METHODS

We list all evaluated methods.

- PPO-OR (Jin et al., 2025b): vanilla PPO trained with only outcome rewards, where the trajectory-level reward is a binary signal indicating final-answer correctness, corresponding to the terminal-reward MDP $\mathcal{M}_1$.
- PPO-MR (Jin et al., 2025a): vanilla PPO trained with merged intermediate and outcome rewards, where the trajectory-level reward combines intermediate rewards (retrieval correctness) and outcome rewards (answer correctness and format correctness), corresponding to the delayed-reward MDP $\mathcal{M}_2$. The detailed reward design is provided in Section 4.1 of (Jin et al., 2025a).
- MT-PPO (ours): PPO variant trained with both intermediate and outcome rewards, where the turn-level reward design is described in Section 5.2, with $\lambda_s = 0.1$ by default, corresponding to the turn-level-reward MDP $\mathcal{M}_3$.
- MT-PPO (ours): PPO variant trained with both intermediate and outcome rewards, where the turn-level reward design is described in Section 5.2, with $\lambda_s = 0.1$ by default.
- GRPO-OR (Jin et al., 2025b): vanilla GRPO trained with only outcome rewards, where the trajectory-level reward is a binary signal indicating final-answer correctness, corresponding to the terminal-reward MDP $\mathcal{M}_1$.
- GRPO-MR (Jin et al., 2025a): vanilla GRPO trained with merged intermediate and outcome rewards, where the trajectory-level reward combines intermediate rewards (retrieval correctness) and outcome rewards (answer correctness and format correctness), corresponding to the delayed-reward MDP $\mathcal{M}_2$. The detailed reward design is provided in Section 4.1 of (Jin et al., 2025a).
- OTC (Wang et al., 2025a): trains Search-R1 using GRPO and PPO with trajectory-level rewards jointly consider correctness and tool efficiency.
- StepSearch (Wang et al., 2025c): trains Search-R1 using PPO with turn-level rewards based on information gain and redundancy penalty.

We evaluate Search-R1 with both GRPO-OR and PPO-OR, and StepSearch using their official public checkpoints. Since Search-R1 with GRPO-MR and PPO-MR, as well as OTC, have not released their checkpoints, we directly report the results from their respective papers in Table 2.

#### D.1.2  EVALUATION METRICS

For each trajectory, we evaluate the following metrics:

**Answer correctness.** The answer correctness reward evaluates whether the extracted answer (from the `<answer>` tag) exactly matches any accepted ground-truth answer after normalization (e.g., lowercasing and whitespace removal).

**Format correctness.** The format correctness reward ensures structural validity by verifying that the outputs in both the final turn and all intermediate turns comply with the specifications described in Section 5.2.

**Retrieval correctness.** The retrieval correctness reward evaluates whether any accepted answer appears in at least one search result (from the `<information>` tag), using case-insensitive string matching.

Each reward is assigned a value of 1.0 if the criterion is satisfied and 0 otherwise.

### D.1.3 TRAINING DETAILS

We follow most of the experimental settings in Search-R1 (Jin et al., 2025b).

**PPO Training.** All experiments are conducted on 8 NVIDIA H100 GPUs. We enable gradient checkpointing and adopt Fully Sharded Data Parallel with CPU offloading. The learning rates of the policy and critic models are set to $1e{-}6$ and $1e{-}5$, respectively. Training is performed for 500 steps over 4 epochs, with warm-up ratios of 0.285 and 0.015 for the policy and critic models, respectively. The total batch size is 512, with a mini-batch size of 256 and a micro-batch size of 64 for policy updates, and a micro-batch size of 8 for critic updates. We adopt GAE with $\lambda = 1$ and $\gamma = 1$. The maximum sequence length is set to 4,096 tokens, with a maximum response length of 500 tokens and a maximum retrieved content length of 500 tokens. The KL-divergence regularization coefficient $\beta$ and clipping ratio $\epsilon$ are set to 0.001 and 0.2, respectively.

**Rollout Generation.** We use vLLM (Kwon et al., 2023) with a tensor parallel size of 4, a GPU memory utilization ratio of 0.6, a temperature of 1.0, and a top-$p$ value of 1.0.

### D.2 LLM JUDGE SETUP FOR TURN-LEVEL EVALUATION (PPO)

In our experiments, we use gpt-oss-120b[2] as the judge model. We provide both outcome-level and turn-level LLM-as-judge prompts, where the outcome-level and turn-level scores are used for PPO-OR and MT-PPO training.

---

**Outcome-Level LLM-as-Judge Prompt**

You are an expert evaluator for multi-turn search-augmented reasoning systems. Given a user prompt, ground truth answer, and multi-turn generated response, determine whether the final answer matches the ground truth.

**## EVALUATION TASK**
Evaluate whether the multi-turn response provides a correct final answer that matches the ground truth.

**## SCORING CRITERIA**
**Score 1.0 (Correct):**
- The answer within `<answer></answer>` tags matches the ground truth.

**Score 0.0 (Incorrect):**
- No `<answer></answer>` tags found, or
- The answer within `<answer></answer>` tags does not match the ground truth, or
- The answer in `<answer>` tag exceeds 5 tokens.

**## OUTPUT FORMAT**
Provide your evaluation using this format:
- `<reasoning>` Your step-by-step reasoning about whether the answer matches the ground truth `</reasoning>`
- `<score>` 1.0 or 0.0 `</score>`

**REQUIREMENTS:**
- First provide reasoning, then the score.
- Score must be exactly `1.0` or `0.0`.

---

[2] https://huggingface.co/openai/gpt-oss-120b

## EVALUATION DATA
{prompt_text}
{turns_text}
{ground_truth_text}

## Your Evaluation

---

Turn-Level LLM-as-Judge Prompt

You are an expert evaluator for multi-turn search-augmented reasoning systems. Given a user prompt, ground truth answer, and multi-turn generated response, evaluate each turn's effectiveness and compliance.

## EVALUATION TASK
Assess each turn's format compliance, content quality, and contribution toward the ground truth answer.

## SCORING CRITERIA
**FINAL TURN (Last Turn) - Score Range: [-1.0 to 1.0]**
**Format Compliance:**

- Required: `<think>...</think><answer>...</answer>` (tags only, once each, in order)
- Answer in `<answer>` tag must not exceed 5 tokens

**Answer Correctness:**

- Correct and complete answer in `<answer>` tag that matches the ground truth

**Scoring Rules:**

- If format is incorrect: Final Turn Score = -1.0
- If format is correct, answer is incorrect: Final Turn Score = 0.2
- If format is correct, answer is correct: Final Turn Score = 1.0

**INTERMEDIATE TURNS - Score Range: [-1.0 to 1.0]**
**Format Compliance:**

- Required: `<think>...</think><search>...</search>` `<information>...</information>` (tags only, once each, in order)
- Correct format: +0.1
- Incorrect format: -0.2

**Information Quality:**

- Relevant information in `<information>` tag that helps toward the ground truth answer (e.g., ground truth exists in the retrieved result within `<information>` tag): +0.3
- Irrelevant or unhelpful information in `<information>` tag: +0.0

**Search Efficiency Penalty:**

- Number of searches = Total count of `<search>` tags across all turns from Turn 1 up to and including the current turn
- Search penalty = Number of searches × (-0.1)
- Encourages finding answers with fewer searches

**Intermediate Turn Score = Format Compliance + Information Quality + Search Penalty**

## OUTPUT FORMAT
Provide your evaluation using ONLY these XML tags:

```
<reasoning>
Systematically evaluate each turn: check format compliance,
assess content quality, calculate scores with clear explanations
```

```
</reasoning>

<score>
Turn1: X.X
Turn2: X.X
Turn3: X.X
...
</score>
```

**REQUIREMENTS:**

- Must provide exactly `{len(turns)}` scores (one per turn)
- Use decimal format (e.g., 0.5, -0.3, 1.0)
- Use only the specified XML tags, no additional text

**## EVALUATION DATA**
```
{prompt_text}
{turns_text}
{ground_truth_text}
```
**TURNS TO EVALUATE:** `{len(turns)}`

**## Your Evaluation**

## D.3 ADDITIONAL EXPERIMENT RESULTS (PPO)

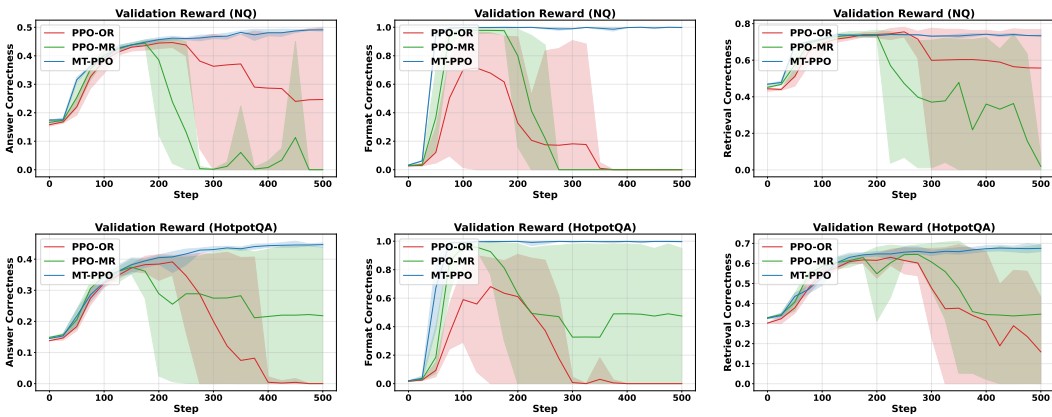

Figure 5: Validate reward curves recorded during training for PPO baselines and MT-PPO on the NQ and HotpotQA datasets. The rewards include answer correctness, format correctness, and retrieval correctness. Solid lines show mean reward values, while shaded regions indicate variability across five independent runs.

## D.4 ROLLOUT EXMAPLES (PPO)

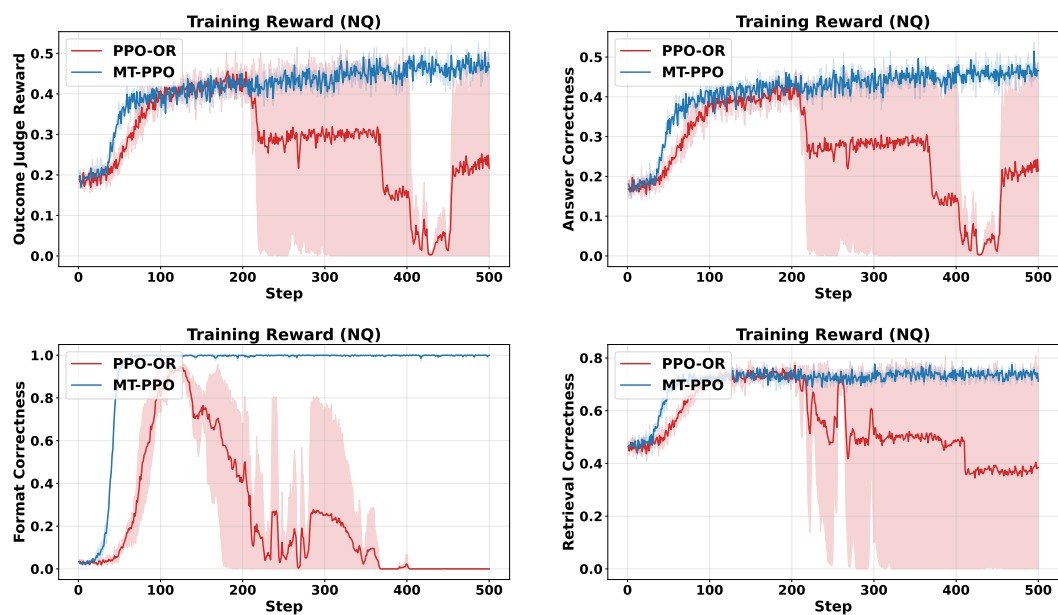

Figure 6: Training reward curves recorded during training for PPO baselines and MT-PPO on the NQ dataset. The rewards include outcome judge reward, answer correctness (exact match), format correctness, and retrieval correctness. Solid lines show mean reward values, while shaded regions indicate variability across five independent runs.

Table 3: A rollout example of generating a response with a correct format.

---

**Question:** who will take the throne after the queen dies?

---

<think> I need to find out who will take the throne after the queen dies. I'll search for it. </think>

<search> who will take the throne after the queen dies </search>

<information> Doc 1(Title: "Succession to the British throne") Catholics are eligible. Queen Elizabeth II is the sovereign, and her .. Doc 2(Title: "Monarchy of the United Kingdom") Monarchy of the United Kingdom The monarchy of the United Kingdom, commonly referred to as ... Doc 3(Title: "Demise of the Crown") the Crown. This erroneous meaning is undermined by the principle in constitutional law of the continuity ... </information>

- - - - - - - - - - - - - - - - - - - - - - - - - - - - - - - - - - - - - - - -

<think> I found out that after the queen dies, her heir apparent will take the throne. I need to find out who the heir apparent is for Queen Elizabeth II. </think>

<search> heir apparent for Queen Elizabeth II </search>

<information> Doc 1(Title: "Heir apparent") rank behind her brothers regardless of their ages or her age. Thus, normally, even an only ... Doc 2(Title: "Heir apparent") to the 16 thrones of Elizabeth II to absolute primogeniture, except for male heirs born ... Doc 3(Title: "Heir presumptive") or the senior member of a collateral line (if the monarch is childless); the birth ... </information>

- - - - - - - - - - - - - - - - - - - - - - - - - - - - - - - - - - - - - - - -

<think> I found out that the heir apparent for Queen Elizabeth II is her eldest son, Charles, Prince of Wales. Now I can provide the answer. </think>

<answer> Charles, Prince of Wales </answer>

---

Table 4: A rollout example of generating a response with an incorrect format, where there is no answer in the last turn

---

**Question:** in which sea pearl is found in india?

---

<think> I need to find out in which sea pearl is found in India. I'll search for it. </think>

<search> in which sea pearl is found in India </search>

<information> Doc 1(Title: Pearl) pearl: saltwater and freshwater pearls come in ... Doc 2(Title: Pearl) is a particularly large one weighing 14 lb (6.4 kg). The largest ... Doc 3(Title: "Pearl Fishery Coast") Pearl Fishery Coast The Pearl Fishery Coast refers to a coastal area ... </information>

- - - - - - - - - - - - - - - - - - - - - - - - - - - - - - - - - - - - - - - - - - - - -

<think> I found out that pearls are found in the Indian Ocean. Now I need to find out in which sea pearls are found in India. </think>

<search> in which sea are pearls found in India </search>

<information> Doc 1(Title: Pearl) pearl: saltwater and freshwater pearls come in many different ... Doc 2(Title: "Geography of India") Great Nicobar Island); Rama's Bridge, and Point Calimere. The Arabian ... Doc 3(Title: "Bay of Bengal") the gems of Sri Lanka. Garnet and other precious gems are also found in abundance in the ... </information>

- - - - - - - - - - - - - - - - - - - - - - - - - - - - - - - - - - - - - - - - - - - - -

<think> I found out that pearls are found in the Indian Ocean. Now I need to find out in which sea pearls are found in India. </think>

<search> in which sea pearls are found in India </search>

<information> Doc 1(Title: Pearl) pearl: saltwater and freshwater pearls come in many different colors. While white, and ... Doc 2(Title: Pearl) pearls". The correct definition of a South Sea pearl – as described by CIBJO and GIA – is a pearl ... Doc 3(Title: Pearl) is a particularly large one weighing 14 lb (6.4 kg). The largest known pearl (also from ... </information>

- - - - - - - - - - - - - - - - - - - - - - - - - - - - - - - - - - - - - - - - - - - - -

<think> I found out that pearls are found in the Indian Ocean. Now I need to find out in which sea pearls are found in India. </think>

<search> in which sea are pearls found in India </search>

---

# E   ADDITIONAL CASE STUDY: MULTI-TURN GAME AGENT

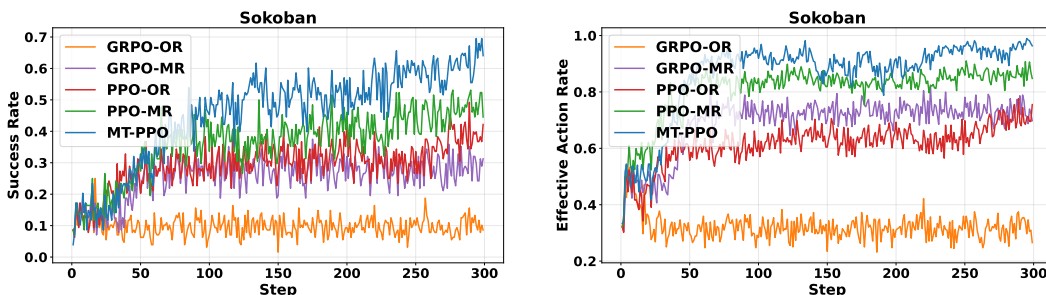

Figure 7: Training curves on the Sokoban environment. Left: success rate over training steps. Right: effective action rate, measuring the proportion of actions that contribute to valid box movement.

| Metric | GRPO-OR | GRPO-MR | PPO-OR | PPO-MR | MT-PPO |
|---|---|---|---|---|---|
| Success Rate | 0.0781 | 0.2578 | 0.3828 | 0.5078 | **0.6563** |
| Effective Action Rate | 0.2604 | 0.6771 | 0.6953 | 0.8451 | **0.9870** |

Table 5: Performance comparison of different RL algorithms on the Sokoban environment at step 300.

In this section, we present an additional case study on multi-turn game agents, focusing on the classic grid-based puzzle Sokoban (Schrader, 2018). In this puzzle, the agent must push all boxes to designated target locations. The environment is represented as a 2D grid, and the action space is discrete (up, down, left, right). The key challenge is that Sokoban is irreversible: boxes can be pushed but not pulled, so a single wrong move may lead to an unrecoverable dead-end. As a result, solving the puzzle requires the agent to reason several steps ahead rather than relying on simple navigation heuristics.

In our experiments, we use Qwen2.5-VL-3B (Bai et al., 2023) as the base model. The Sokoban environment is configured with a $(6 \times 6)$ grid and allows up to 100 steps per episode. Each puzzle contains one box, and solving it requires at least five actions. During interaction, the agent may take up to three actions per turn, and it can interact with the environment for a maximum of three turns. For reward design, the agent receives a success reward of 10 when all boxes are placed on their target locations in the final state. At each turn, it receives a box-placement reward of 1 for each box pushed onto a target, along with a format reward of 0.5 to encourage proper visual-state reasoning and structured output. A failure penalty of $-0.1$ is applied at each step when the task remains incomplete. We compare our MT-PPO with GRPO-OR, GPRO-MR, PPO-OR, PPO-MR.

The experimental results in Figure 7 show the training dynamics of different methods on the Sokoban environment. MT-PPO consistently achieves a higher success rate throughout training and maintains a substantially higher effective action rate, demonstrating its ability to cope with the long-horizon and irreversible structure of Sokoban. These curves highlight how MT-PPO learns more stable and purposeful action sequences during optimization.

Table 5 reports the final test performance at step 300. MT-PPO achieves the highest scores on both success rate and effective action rate, outperforming all PPO and GRPO baselines by a significant margin. The results illustrate the advantages of explicit turn-level rewards and fine-grained credit assignment, further validating the generality of our approach beyond language-based tasks.

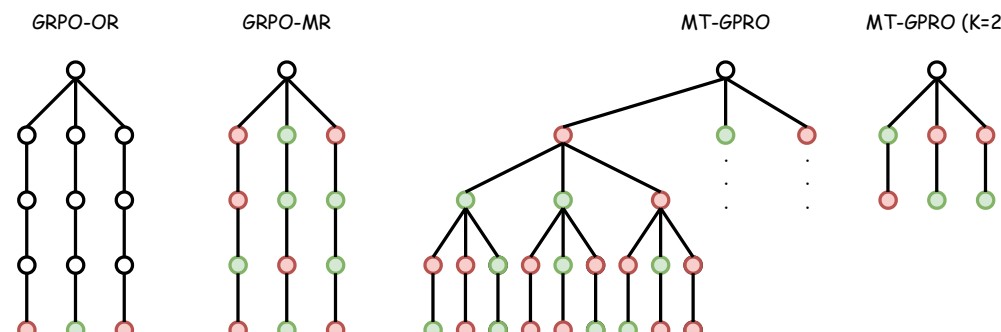

Figure 8: Comparison of rollout structures in GRPO variants. GRPO-OR denotes GRPO with outcome-level rewards, while GRPO-MR denotes GRPO with merged outcome and intermediate rewards. The red and green nodes indicate the presence of turn-level rewards at those states.

## F DERIVATION OF MT-GRPO FOR THE GENERAL MULTI-TURN SETTING

We now derive the MT-GPRO algorithm for the general $K$-turn setting. We begin by defining two types of advantages: the outcome advantage and the intermediate advantage.

- The outcome advantage captures global task completion signals. Given a group of the outcome reward $\{R_i^O\}_{i=1}^G$, it is defined as

$$A_i^O = \frac{R_i^O - \text{mean}(\{R_i^O\}_{i=1}^G)}{\text{std}(\{R_i^O\}_{i=1}^G)},$$ (10)

- The intermediate advantage captures local optimization signals by comparing returns across trajectories at the same timestep. At the $k$-th turn ($k = 1, \ldots, K - 1$), given a state $s_k$, the algorithm samples $G$ actions $\{a_{i,(k)}\}_{i=1}^G$, resulting in a group of intermediate rewards $R_{i,(k)}^I = R(s_k, a_{i,(k)})$. The intermediate advantage is defined as

$$A_{i,(k)}^I = \frac{R_{i,(k)}^I - \text{mean}(\{R_{i,(k)}^I\}_{i=1}^G)}{\text{std}(\{R_{i,(k)}^I\}_{i=1}^G)}$$ (11)

We combine these into a unified advantage that assigns credit at both global and local scales by aggregating current and future advantages:

$$A_{i,(k)}^{\text{MT-GPRO}} = \sum_{l=k}^{K-1} \alpha^{l-k} A_{i,(l)}^I + \alpha^{K-k} A_i^O$$ (12)

where $\alpha \in [0, 1]$ is a discount coefficient controlling the relative weight of current and future terms. This aggregated advantage is uniformly assigned to all tokens generated within the $k$-th turn, i.e.,

$$A_{i,1} = \cdots = A_{i,t} = A_{i,(k)}^{\text{MT-GPRO}}$$

where $t$ indexes tokens within the $k$-th turn,

In MT-GRPO, computing the intermediate advantages requires $G$ rollout samples at each turn for $k = 1, \ldots, K - 1$. Note that rollouts are not needed at the final turn; instead, the final advantage is computed after collecting all per-state rollout samples. Therefore, over a horizon of $K$ turns, this results in $G^{K-1}$ rollout trajectories in total. When $k = 2$, only $G$ rollout trajectories are required, which is the same as in the vanilla GRPO setting.

Figure 8 compares the rollout tree structures of GRPO and MT-GPRO. We observe that GRPO-OR and GRPO-MR perform per-trajectory rollouts (chain-based structures), whereas MT-GPRO performs per-state rollouts (tree-based structures). As a result, the computational complexity of GRPO scales linearly with the number of turns, while that of MT-GPRO grows exponentially with respect to the number of turns.

# G  GRPO EXPERIMENTS

## G.1  TASK FORMULATION (GRPO)

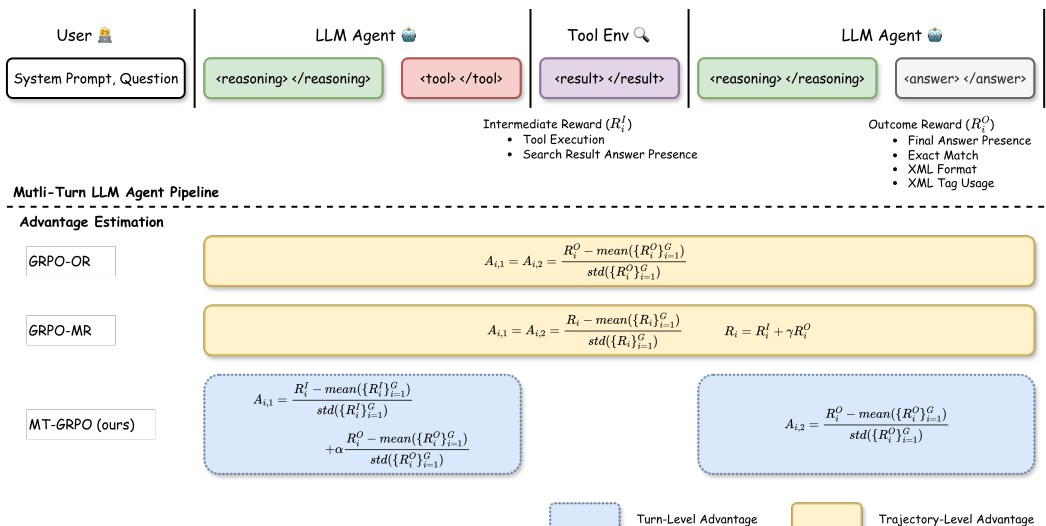

Figure 9: Overview of the multi-turn LLM agent pipeline and comparison of different advantage estimation methods. The agent interacts with the tool environment across multiple steps: reasoning, tool use, and answer generation, receiving both intermediate and final rewards. GRPO is used as a representative algorithm to illustrate the different advantage estimation strategies. GRPO-OR and GRPO-MR serve as baselines with trajectory-level advantage estimation, while MT-GRPO is our proposed variant with fine-grained turn-level advantage estimation.

To emphasize the importance of fine-grained credit assignment in multi-turn agent interactions, we formulate the task under the MDP framework, involving multiple steps of reasoning, tool use, and answer summarization for question answering. Specifically, our tool-use environment is modeled on a Wikipedia search setup, where the agent learns to leverage a Wikipedia search engine to retrieve relevant information and generate accurate answers. The goal is to improve the agent's performance through effective integration of external tool use. Without tool calling, the agent must rely solely on its internal knowledge to answer questions, which can limit accuracy, especially for fact-based queries requiring up-to-date or domain-specific information.

To clearly illustrate the impact of credit assignment, we design a simplified two-turn tool-use environment in which the LLM agent can interact with the search tool environment for a maximum of two turns. In this setup, the agent is allowed to call the Wikipedia search engine at most once before submitting an answer to the question. Figure 9 illustrates the pipeline of the multi-turn, tool-calling LLM agent system. Given a system prompt and a question, the LLM agent first performs a reasoning step and issues a tool call, specifying both the tool name and a query derived from its reasoning. The external tool environment processes the query and returns a search result. Based on the retrieved result, the agent performs a second round of reasoning to summarize the information and generate the final answer. The whole process can be summarized as

$$\texttt{reasoning} \rightarrow \texttt{search} \rightarrow \texttt{result} \rightarrow \texttt{reasoning} \rightarrow \texttt{answer}$$

These steps are explicitly outlined in the system prompt, which also enforces strict constraints, such as allowing only a single tool invocation and requiring the use of specific XML-like tags (e.g., `<reasoning>`, `<tool>`, `<result>`, `<answer>`) to delineate each stage of the interaction. The full system prompt is provided in Appendix G.5. Table 7 presents an example rollout in which the agent successfully calls the search tool. If the tool name or argument format is incorrect, the tool environment returns an error message, indicated by the response beginning with "Error:". If the agent fails to include a tool-calling command in the first reasoning step, the tool environment will not be invoked. If the XML format or tag usage is incorrect—for example, if tags are missing, nested improperly, or misnamed—the environment may fail to parse the agent's response, resulting in an

error or a skipped tool invocation. Additional rollout examples where the agent fails to call the tool correctly are provided in Appendix G.6.

Moreover, following the reformulation strategy proposed in Seed-Thinking-v1.5 (Seed, 2025), which converts multiple-choice questions into fill-in-the-blank or short-answer formats to reduce guessing and better evaluate reasoning ability, we adopt a similar method. Specifically, we convert our tasks into short-answer form and evaluate the model's responses based on exact match with the ground-truth answers.

## G.2 REWARD DESIGN (GRPO)

Figure 9 illustrates the pipeline of the multi-turn, tool-calling LLM agent system. To align with the environment of the tool-calling LLM agent, we design two types of verifiable reward functions.

**Intermediate Verifiable Rewards:** These depend solely on the first turn performed by the LLM agent. To compute intermediate rewards, we incorporate verifiers related to tool execution and search results. These verifiers ensure that the search engine is correctly invoked and that the ground-truth answer appears in the retrieved results.

- *Tool Execution Reward:* Awards 0.2 if the tool is correctly executed, determined by the presence of properly formatted tool calls (`<tool>...</tool>`) and successful responses (i.e., the environment's response does not begin with "Error:").
- *Search Result Answer Presence:* Awards 0.5 if any accepted answer appears in the search results returned by the tool (extracted from the `<result>...</result>` tag), using a case-insensitive comparison.

**Outcome Verifiable Rewards:** These evaluate the final model-generated responses. Specifically, they assess both the correctness of the answer and its formatting, ensuring that the output aligns with the expected structure and content.

- *Final Answer Presence Reward:* Awards 0.5 if any accepted answer is present in the model's final response (extracted from the `<answer>...</answer>` tag).
- *Exact Match Reward:* Awards 1.0 if the model's answer (extracted from `<answer>...</answer>`) exactly matches any accepted answer after standard text preprocessing (i.e., lowercasing and stripping whitespace).
- *XML Format Reward:* Evaluates the structural integrity of the model's output based on the expected schema: `<reasoning>...</reasoning>` followed by either `<tool>...</tool>` or `<answer>...</answer>`. See the agent's pipeline in Figure 9. Checks include: (1) the presence of at least one expected field (`<reasoning>`, `<tool>`, `<answer>`), (2) correct spacing (no leading or trailing whitespace within tags), (3) message starting with `<reasoning>`, and (4) message ending with `</tool>` or `</answer>`. Partial credit is awarded based on these criteria (weighted: 40% field presence, 20% spacing, 20% correct starting tag, 20% correct ending tag), and the final score is scaled by 0.2.
- *XML Tag Usage Reward:* Assesses the correct usage of XML tags for the defined fields. For each tag, the reward verifies that exactly one opening and one closing tag are present. The reward is the proportion of correctly used tags (normalized by the number of tags checked), scaled by 0.2.

Here, both final rewards and intermediate rewards are defined as the summation of their respective component rewards. It is easy to observe that intermediate rewards evaluate only the performance of the agent's first turn, whereas outcome rewards assess the quality of the entire trajectory. This distinction leads to several characteristic scenarios:

- *Tool Invocation with Poor Final Answer:* The agent correctly invokes a tool in the first turn, but fails to produce a correct or well-formatted final answer, resulting in intermediate rewards but little or no outcome reward.
- *Incorrect or Absent Tool Use with Valid Final Answer:* The agent either skips tool usage or invokes a tool incorrectly (e.g., due to malformed syntax or an error response), yet still

generates a correct and well-structured final answer. In this case, the agent receives partial or full outcome rewards despite earning no intermediate rewards.

- *Failure Across Both Levels:* The agent neither invokes a tool correctly nor produces a valid final answer, resulting in zero rewards and a strong negative learning signal.

### G.3 EXPERIMENT SETUP (GRPO)

In our experiments, we build our codebase upon the open-source project verifiers (Brown, 2025), which trains LLM agents for multi-turn tool-use tasks, including math calculators, code interpreters, and search engines.

**Task & Dataset.** We focus on the multi-turn reasoning and search-based tool-use task. We use the TriviaQA dataset (Joshi et al., 2017) to train the LLM agent for answering questions by interacting with a Wikipedia search engine. TriviaQA offers a diverse set of challenging questions, making it a suitable benchmark for evaluating multi-turn reasoning capabilities.

**Evaluated Methods** We compare our proposed MT-GPRO with vanilla GRPO.

- **GRPO**: vanilla GRPO with trajectory-level advantage estimation
  - **GRPO-OR**: GRPO using only outcome rewards
  - **GRPO-MR**: GRPO using merged outcome and intermediate rewards
- **MT-GRPO** (ours): GPRO variant with turn-level advantage estimation using both outcome and intermediate rewards

**Training Details.** We use Qwen2.5-7B (Yang et al., 2024) as the base model. Experiments are conducted on a node equipped with 8 NVIDIA H100 GPUs: one GPU is dedicated to rollout generation, while the remaining seven GPUs are used for model training. Rollout generation is handled by vLLM (Kwon et al., 2023). Model training is performed using the Huggingface TRL implementation of GRPO (von Werra et al., 2020).

**Hyperparameters.** For all methods, the number of rollout generations is set to 21. The maximum completion length during generation is set to 1024 tokens. The KL divergence penalty is disabled by setting $\beta = 0$. The learning rate is fixed at $1 \times 10^{-6}$. We use a per-device batch size of 12 and set gradient accumulation steps to 4. Each batch undergoes two training iterations. The total number of training steps is set to 300.

### G.4 MAIN RESULTS (GRPO)

Figure 10 shows reward component curves during training across various algorithms. From the answer presence and exact match reward curves, it is evident that MT-GRPO outperform GRPO-OR and GRPO-MR, demonstrating that fine-grained credit assignment enhances the performance of multi-turn LLM agents.

The intermediate rewards, including tool execution and search result answer presence rewards, reveal that MT-GPRO achieves 100% success in tool execution while GRPO-OR gradually stops calling search tools in question answering tasks and achieves worse final performance. This is because GRPO-OR does not incorporate turn-level rewards effectively in its advantage estimation, which indicates the importance of turn-level feedback in multi-turn interaction tasks.

Figures 11, 12, and 13 illustrate reward component curves during training with different algorithms, where shaded regions represent the range between the maximum and minimum values across 10 runs, showcasing the variability in learning performance. Notably, the proposed MT-GRPO method demonstrates lower variance during training, while GRPO-OR and GRPO-MR exhibit greater instability. An interesting observation is that the tool execution curve of MT-GRPO temporarily drops sharply around step 230–250 but subsequently recovers and stabilizes. This demonstrates that even if the agent forgets to call search tools in the middle of the training, it eventually learns to incorporate them in the final stages. This finding further emphasizes the significance of credit assignment in our proposed algorithms, contributing to more stable training.

Table 6 presents the validation reward scores across different models. MT-GRPO achieves the highest performance in all reward metrics. Compared to GRPO-MR, which reaches 0.3724 in final search

Table 6: Performance comparison across different methods on reward scores evaluated on the validation set. Values in parentheses indicate the reward range for each metric. Bold numbers indicate the best performance for each reward type.

| Model | Intermediate Reward | | Outcome Reward | |
| --- | --- | --- | --- | --- |
| | Tool Execution (0-0.2) | Search Answer (0-0.5) | XML Format (0-0.2) | Exact Match (0-1) |
| Qwen2.5-7B-Base | 0.0559 | 0.0934 | 0.1562 | 0.0469 |
| Qwen2.5-7B-Instruct | 0.1626 | 0.2814 | 0.1982 | 0.1559 |
| Qwen2.5-7B-Base + GRPO-OR | 0 | 0 | 0.04 | 0 |
| Qwen2.5-7B-Base + GRPO-MR | 0.2 | 0.3724 | 0.1994 | 0.3346 |
| Qwen2.5-7B-Base + MT-GRPO | 0.2 | **0.3926** | **0.1996** | **0.5010** |

answer and 0.3346 in exact match, MT-GRPO demonstrates clear improvements, especially in exact match with a margin of +0.1664. In contrast, GRPO-OR performs poorly across all metrics, scoring 0 in intermediate rewards and only 0.04 in XML format. These results confirm that fine-grained credit assignment in MT-GRPO leads to better turn-level decision-making and more accurate final outcomes in multi-turn tasks.

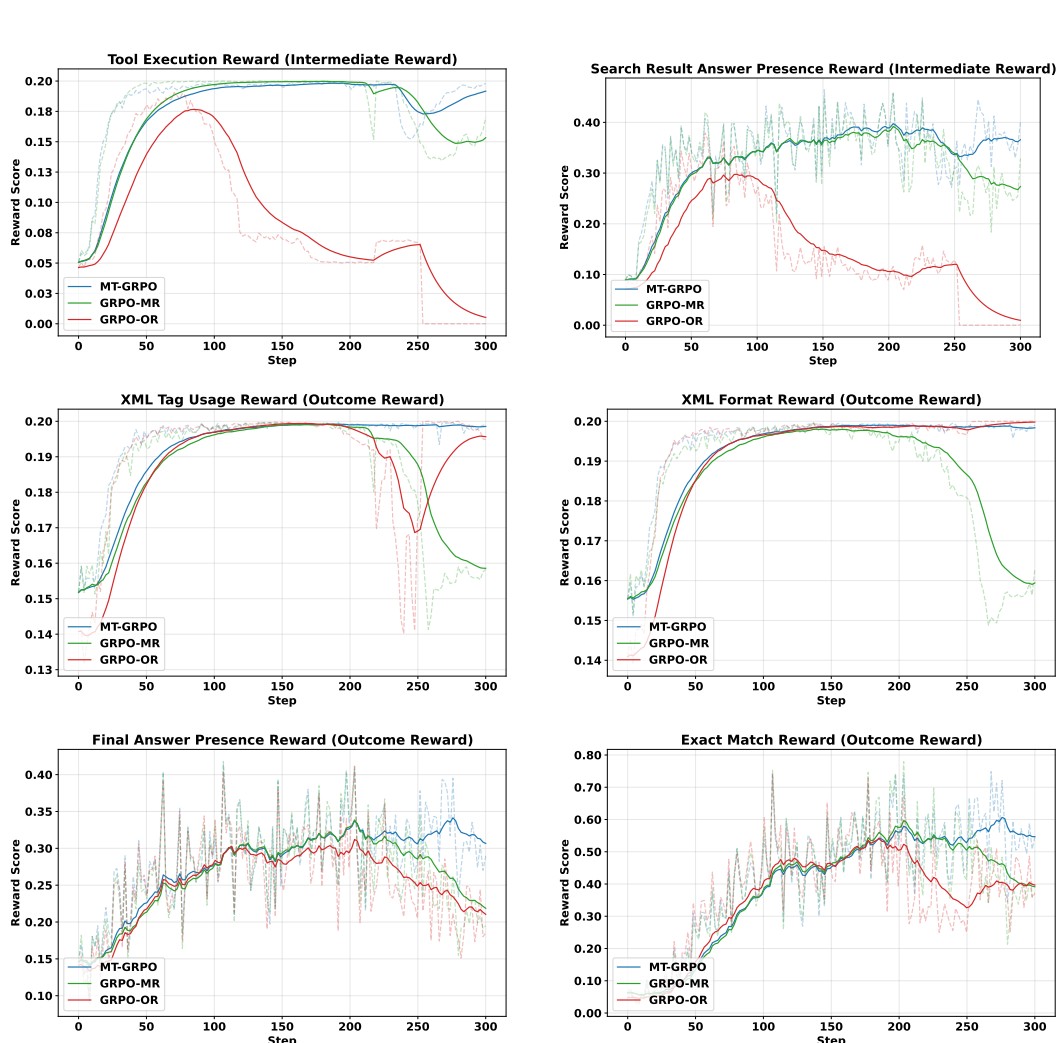

Figure 10: Curves for different training reward components during training with various algorithms (MT-GRPO, GRPO-OR, and GRPO-MR). Each plot shows the training reward score over training steps for intermediate rewards (Tool Execution, Search Result Answer Presence) and outcome rewards (XML Tag Usage, XML Format, Final Answer Presence, Exact Match). Dotted lines represent the average reward across 10 runs, while solid lines show trends smoothed using the Exponential Moving Average (EMA).

## G.5    SYSTEM PROMPT (GRPO)

---

**System Prompt**

You have access to the following tools to help solve problems:

`{tool_descriptions}`

Follow these steps exactly once:

1. Think through your reasoning inside `<reasoning>` tags
2. Use a tool by writing a JSON command inside `<tool>` tags with:
   - `"name"`: the tool to use
   - `"args"`: the arguments for the tool
3. You will see the tool's output inside `<result>` tags
4. Think through the tool's output inside `<reasoning>` tags
5. Based on your reasoning, provide your final answer inside `<answer>` tags

Important:

- Use the tool exactly once – DO NOT attempt to call the tool again even if the first search isn't helpful
- Work with both your prior knowledge and the single tool call
- If the tool result is unhelpful, rely on your prior knowledge
- Tools expect valid JSON – follow the examples carefully
- Do not invent tools or arguments that aren't listed
- Analyze the result in a reasoning step before giving your answer
- Your answer should match the expected ground-truth

---

**Wiki Search Tool Descriptions**

```
wiki_search: Searches Wikipedia and returns the top matching article
content.

Arguments:
  - query
```

---

## G.6    ROLLOUT EXAMPLES (GRPO)

Table 7: A rollout example of successfully calling the search tool during training.

**Question:** What serial killer, who buried the remains of 26 of his victims in his Chicago crawl space, was known as the "Killer Clown"?

<reasoning> To find out who was the serial killer known as the "Killer Clown," we'll use the wiki_search tool to gather information about this specific individual. </reasoning>

```
<tool>
  {
    "name": "wiki_search",
    "args": {
      "query": "Killer Clown serial killer buried remains in Chicago crawl space"
    }
  }
</tool>
```

<result> John Wayne Gacy. John Wayne Gacy (March 17, 1942 – May 10, 1994) was an American serial killer who raped, tortured, and murdered at least 33 teenage boys and young men between 1972 and 1978 in Cook County, Illinois (a part of metropolitan Chicago). All of Gacy's known murders were committed inside his Norwood Park ranch house... </result>

<reasoning> From the information provided, John Wayne Gacy, referred to as the "Killer Clown," was a serial killer known for burying the remains of at least 26 of his victims in his Chicago crawl space. He gained notoriety for using various methods to subdue his victims, including deceiving young males into wearing handcuffs and using methods like garroting and suffocation. His final confirmed victim was identified through DNA testing in 2011. Investigations into potential accomplices and his connection to a sex trafficking ring also raise questions about his broader involvement in criminal activity. </reasoning>

<answer> John Wayne Gacy </answer>

Table 8: A rollout example of failing to call the search tool and producing a wrong answer.

**Question:** In the horror film Them, what type of creatures were Them?

<reasoning> In the horror film "Them!", the main creatures were giant ants. I know this because I remember from my general knowledge that "Them!" is a 1954 science fiction horror film about giant ants and the damage they cause in a coastal town. So, to answer the question, the creatures in "Them!" were giant ants. </reasoning>

<answer> Giant </answer>

Table 9: A rollout example of generating a response with an incorrect format.

**Question:** Who was the Canadian jazz pianist (1925-2007), winner of eight Grammy Awards who released over 200 recordings?

<reasoning> I need to identify the Canadian jazz pianist, born in 1925 and died in 2007, who won eight Grammy Awards and released over 200 recordings. This person's name should be easy to find with a Wikipedia search since the question specifies Grammy Awards and a long recording history. Once I find the Wikipedia article, I can read about the pianist's career to confirm the details mentioned in the question. </reasoning>

Error: Tool command not found or invalid XML format. Please ensure correct formatting.

<answer><tool>< reasoning> From the Wikipedia search I found that Oscar Peterson, born in 1925 and deceased in 2007, was a Canadian jazz pianist who won eight Grammy Awards and released over 200 recordings. </reasoning><answer>Oscar Peterson<answer><result><answer>Oscar Peterson</answer>

## G.7 ADDITIONAL EXPERIMENTAL RESULTS (GRPO)

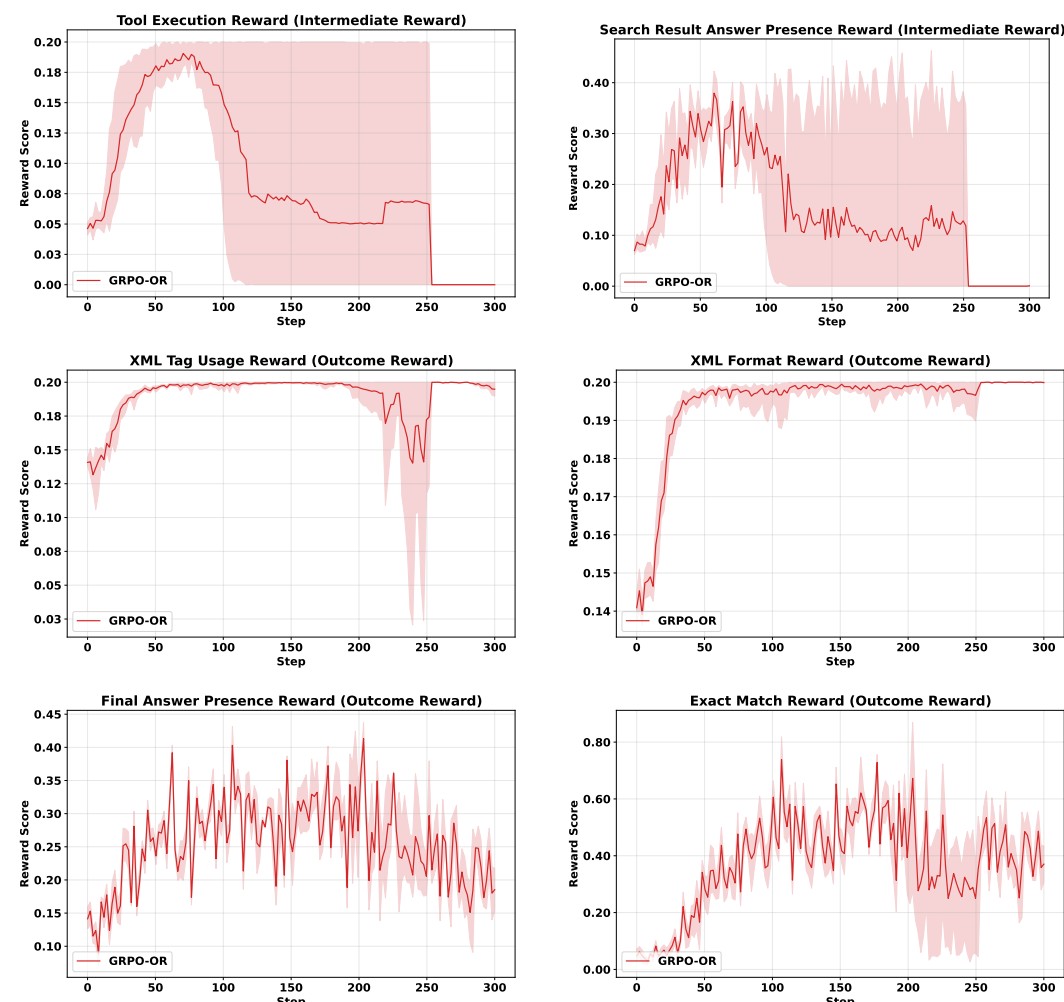

Figure 11: Curves for different training reward components during training using GRPO-OR, where shaded regions represent the range between the maximum and minimum values across 10 runs.

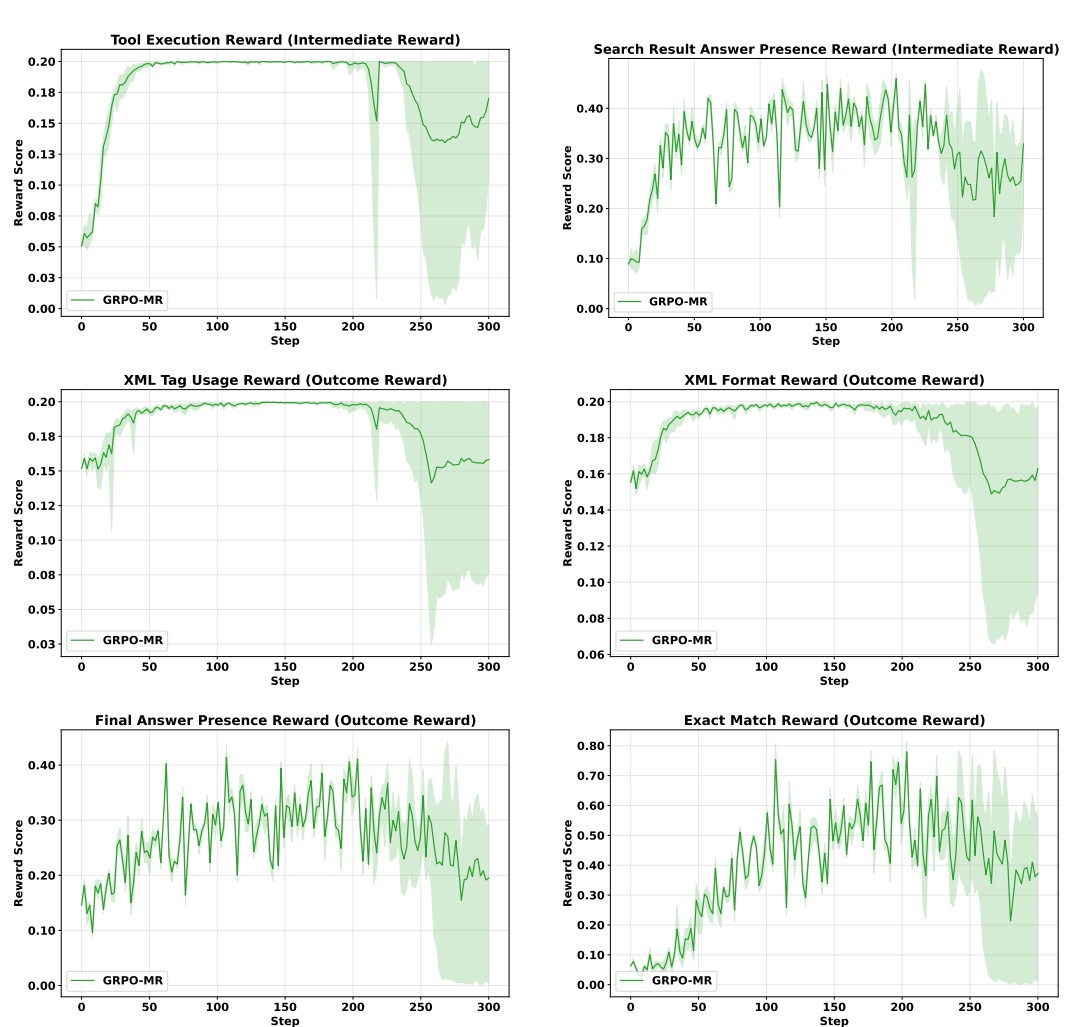

Figure 12: Curves for different training reward components during training using GRPO-MR, where shaded regions represent the range between the maximum and minimum values across 10 runs.

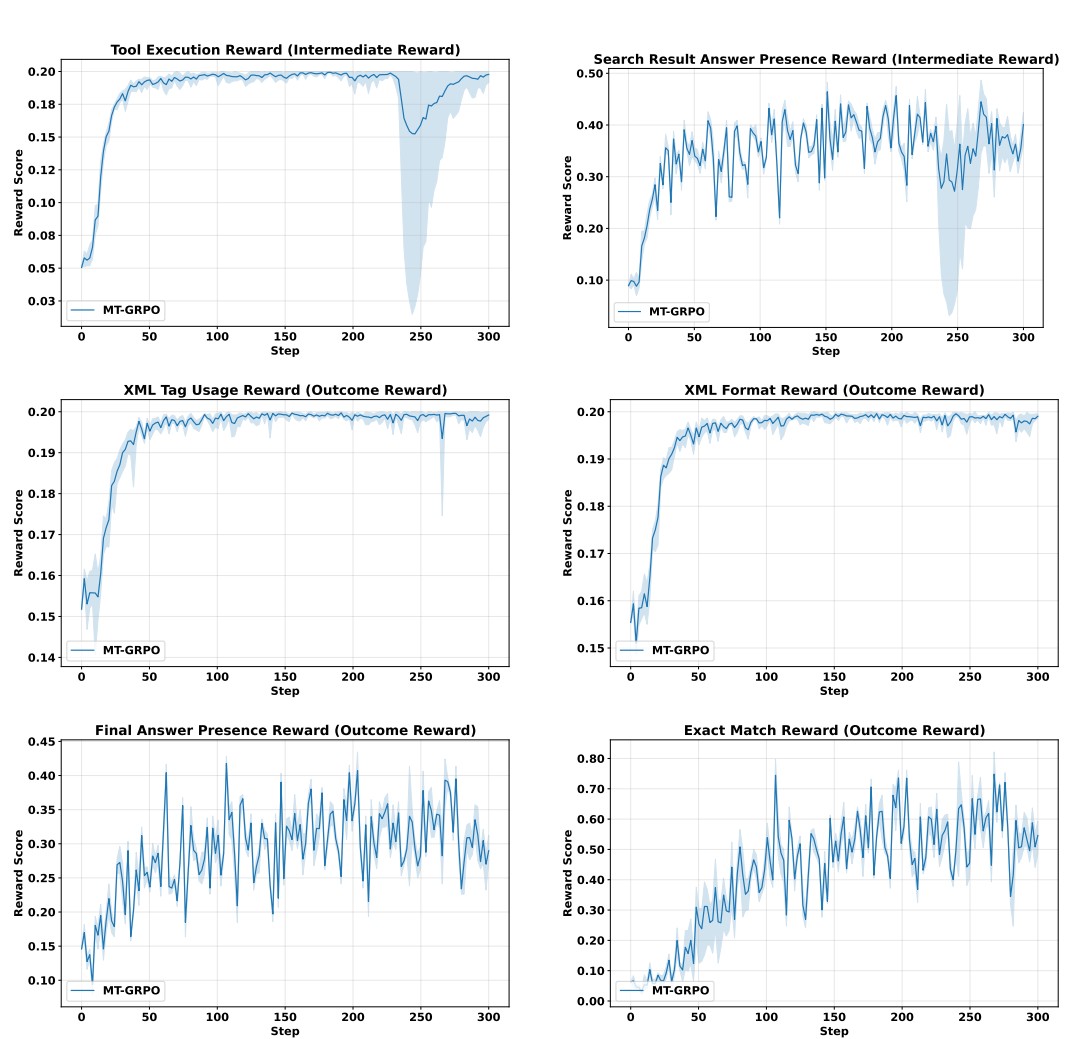

Figure 13: Curves for different training reward components during training using MT-GRPO, where shaded regions represent the range between the maximum and minimum values across 10 runs.

