# OpenReview forum: "Reinforcing Multi-Turn Reasoning in LLM Agents via Turn-Level Reward Design"
_ICLR.cc/2026/Conference — Submitted to ICLR 2026_

### Official Review · Reviewer_JWKY · 2025-10-26

**Soundness:** 3
**Presentation:** 3
**Contribution:** 2
**Rating:** 4
**Confidence:** 3

**Summary:**

They propose a turn-level reward design strategy to enhance RL algorithms in multi-turn agent tasks. By integrating turn-level rewards, they extend GRPO and PPO to their respective multi-turn variants, enabling fine-grained credit assignment.

**Strengths:**

- The algorithm is well clarified with a specific case study.
- This paper studies a fundamental problem for multi-turn RL -- the use of turn-level reward.
- The MT versions of PPO and GRPO show better performance compared to their counterparts: PPO and GRPO.

**Weaknesses:**

- Lack of theoretical support.
- Limited Baselines for Comparison: To provide a more comprehensive evaluation, additional baselines should be included, such as GRPO or PPO augmented with intrinsic rewards. The current comparisons are restricted to open-source LLMs and ablated variants of the algorithm, which may not fully benchmark the approach against state-of-the-art reinforcement learning methods in similar domains.
- Omission of Concurrent Works: The discussion should address relevant concurrent research, such as the work on "Context-lite Multi-turn Reinforcement Learning for LLM Agents," to highlight how the proposed method differentiates itself or builds upon these efforts.

**Questions:**

See weaknesses.

---

> ### Author Response · Authors · 2025-11-21
>
> > W1. Lack of theoretical support
>
> **Response**: We thank the reviewer for raising this point. We agree that theoretical support helps improve our paper. We have revised Section 2 to more clearly compare different MDP formulations and have added new theoretical results in Appendix C that formally compare the policy-gradient variance of different MDP formulations. Our new theoretical results highlight the importance of turn-level rewards.
>
> Based on the granularity of reward assignment, we categorize multi-turn formulations into three types of turn-level MDPs:
> * Turn-level MDP with a terminal reward $\mathcal{M}_1$: provides only a final outcome (terminal) reward with no intermediate rewards.
> * Turn-level MDP with a delayed reward $\mathcal{M}_2$: provides an accumulated reward that merges both intermediate and outcome rewards into a single delayed signal.
> * Turn-level MDP with explicit turn-level rewards $\mathcal{M}_3$: provides explicit rewards at each turn.
>
> Here, $\mathcal{M}_1$ contains only outcome rewards, whereas both $\mathcal{M}_2$ and $\mathcal{M}_3$ include intermediate rewards but differ in how these rewards are distributed across the turns. Moreover, $\mathcal{M}_1$ and $\mathcal{M}_2$ provide trajectory-level rewards, whereas $\mathcal{M}_3$ provides explicit turn-level rewards. Notably, most existing multi-turn agent studies adopt either $\mathcal{M}_1$ or $\mathcal{M}_2$. In contrast, our paper focuses on $\mathcal{M}_3$.
>
> We analyze and compare the policy-gradient variance of the two return-equivalent MDPs $\mathcal{M}_2$ and $\mathcal{M}_3$, since the comparison between $\mathcal{M}_1$ and $\mathcal{M}_3$ is trivial. Although $\mathcal{M}_2$ and $\mathcal{M}_3$ share the same optimal $Q$-values and thus the same optimal policies, they differ in how rewards are assigned across turns. Our theoretical results show that the MDP with explicit turn-level rewards $\mathcal{M}_3$ yields lower policy-gradient variance than the MDP that relies solely on a delayed accumulated reward $\mathcal{M}_2$. These findings help justify our formulation and highlight why turn-level rewards and fine-credit credit assignment leads to more stable and efficient RL training for GRPO and PPO.
>
> We hope that this expanded theoretical treatment adequately addresses the reviewer’s concern and strengthens the conceptual contributions of the paper.

---

> > ### Author Response · Authors · 2025-11-21
> >
> > > W2. Limited Baselines for Comparison: To provide a more comprehensive evaluation, additional baselines should be included, such as GRPO or PPO augmented with intrinsic rewards. The current comparisons are restricted to open-source LLMs and ablated variants of the algorithm, which may not fully benchmark the approach against state-of-the-art reinforcement learning methods in similar domains.
> >
> >
> > **Response**: We appreciate the reviewer’s suggestion. We agree that additional baselines should be included to provid a more comprehensive evaluation. In the revised manuscript, we have expanded our experimental comparisons to include several additional state-of-the-art RL baselines in this domain.
> >
> > For training dynamics, we include PPO-MR from the original Search-R1 series \[1\], vanilla PPO trained with merged intermediate and outcome rewards, where the trajectory-level reward jointly captures retrieval correctness, answer correctness, and format correctness. For benchmark performance, we additionally include state-of-the-art RL baselines, OTC \[2\] and StepSearch \[3\]. OTC trains Search-R1 using GRPO and PPO with trajectory-level rewards jointly consider correctness and tool efficiency at each turn. StepSeatch trains Search-R1 using PPO with turn-level rewards based on information gain and redundancy penalty. As shown in the table below, with full results provided in Table 2 and Figure 3 in Section 6, MT-PPO outperforms all evaluated baselines on all datasets. We hope this addresses the reviewer's concern.
> >
> >
> > | Methods                 | NQ    | TriviaQA | PopQA | HotpotQA | 2wiki | Musique | Avg.  |
> > |-------------------------|-------|----------|-------|----------|-------|---------|-------|
> > | **Qwen2.5-7B-Base**     | 0.177 | 0.319    | 0.181 | 0.160    | 0.167 | 0.040   | 0.174 |
> > | **Qwen2.5-7B-Instruct** | 0.320 | 0.563    | 0.349 | 0.292    | 0.277 | 0.118   | 0.320 |
> > | **GRPO-OR (Search-R1)** | 0.391 | 0.560    | 0.388 | 0.331    | 0.306 | 0.129   | 0.351 |
> > | **GRPO-MR (Search-R1)** | 0.453 | 0.628    | 0.450 | 0.416    | 0.375 | 0.164   | 0.414 |
> > | **PPO-OR (Search-R1)**  | 0.483 | 0.639    | 0.456 | 0.435    | 0.382 | 0.199   | 0.432 |
> > | **PPO-MR (Search-R1)**  | 0.472 | 0.629    | 0.452 | 0.436    | 0.402 | 0.180   | 0.429 |
> > | **GRPO (OTC)**          | 0.444 | 0.597    | 0.431 | 0.366    | 0.311 | 0.130   | 0.380 |
> > | **PPO (OTC)**           | 0.446 | 0.623    | 0.425 | 0.383    | 0.363 | 0.152   | 0.399 |
> > | **PPO (StepSearch)**    | 0.355 | 0.570    | 0.385 | 0.351    | 0.396 | 0.179   | 0.373 |
> > | **MT-PPO (ours)**       | **0.490** | **0.647** | **0.459** | **0.453** | **0.424** | **0.209** | **0.447** |
> >
> >
> > \[1\] Bowen Jin, Jinsung Yoon, Priyanka Kargupta, Sercan O. Arik, and Jiawei Han. _An empirical study on reinforcement learning for reasoning-search interleaved LLM agents._ arXiv:2505.15117, 2025.
> > \[2\] Hongru Wang, Cheng Qian, Wanjun Zhong, Xiusi Chen, Jiahao Qiu, Shijue Huang, Bowen Jin, Mengdi Wang, Kam-Fai Wong, and Heng Ji. _OTC: Optimal tool calls via reinforcement learning._ arXiv:2504.14870, 2025.
> > \[3\] Ziliang Wang, Xuhui Zheng, Kang An, Cijun Ouyang, Jialu Cai, Yuhang Wang, and Yichao Wu. _StepSearch: Igniting LLMs’ search ability via step-wise proximal policy optimization._ arXiv:2505.15107, 2025.

---

> > > ### Author Response · Authors · 2025-11-21
> > >
> > > > W3. Omission of Concurrent Works: The discussion should address relevant concurrent research, such as the work on "Context-lite Multi-turn Reinforcement Learning for LLM Agents," to highlight how the proposed method differentiates itself or builds upon these efforts.
> > >
> > >
> > > **Response**: Thank the reviewer for highlighting this relevant concurrent work. The referenced paper is indeed strong work and we are happy to discuss its relationship to our method.
> > >
> > > It proposes a dual-discounting GAE approach that decouples step-level and token-level credit assignment. While related, the key difference from our work is that they rely on value-based credit assignment at the step level, whereas our method directly incorporates explicit intermediate rewards into the PPO objective. We have added a discussion of this work in the Related Work section (Section B) to better situate our contributions in the evolving research landscape. We also plan to include empirical comparisons once the authors release their code. We hope this addresses the reviewer's concern.

---

### Official Review · Reviewer_SxgQ · 2025-10-29

**Soundness:** 2
**Presentation:** 3
**Contribution:** 1
**Rating:** 2
**Confidence:** 4

**Summary:**

This paper proposes a turn-level reward design framework to improve reinforcement learning for multi-turn LLM agents. The authors extend GRPO and PPO into multi-turn variants (MT-GRPO and MT-PPO) that integrate intermediate rewards to enable finer credit assignment across reasoning steps. They evaluate the approach on search-based QA tasks using both verifiable and LLM-as-judge rewards. Experimental results with Qwen2.5-7B show that MT-PPO achieves more stable training, faster convergence, and better format correctness than PPO and GRPO. The paper highlights turn-level reward design as a promising direction for long-horizon agent training.

**Strengths:**

1. The paper tackles an important problem: improving multi-turn reasoning in LLM agents through better reward shaping
2. The distinction between single-turn and multi-turn MDP formulations is well presented and conceptually sound.
3. The paper is clearly written and easy to follow, with consistent notation and illustrative examples.

**Weaknesses:**

1. The main contribution, introducing turn-level rewards into PPO/GRPO, is conceptually straightforward and closely related to prior work on process reward models (PRM) and segment-level credit assignment. The paper overstates its originality by claiming to be the “first systematic study” without adequately discussing/comparing with these prior methods.
2. The experiments are limited to search-based QA tasks, leaving it unclear whether the proposed framework generalizes to other multi-turn or open-ended domains such as code generation, dialogue, or planning.
3. The reported improvement in answer accuracy (approximately +1.5% over PPO) is relatively modest compared to the additional implementation complexity required for designing and tuning intermediate rewards.
4. The motivation for introducing MT-GRPO is weakly justified, and its evaluation is minimal.
5. Novelty concern: Much of the paper’s content overlaps with existing work, and the contributions are largely incremental.

**Questions:**

1. How does MT-PPO performance compare to prior PRM or step-level RL work?
2. How sensitive are results to the chosen reward weights (retrieval, format, search penalty)?
3. What is the computational overhead of MT-PPO relative to standard PPO (in terms of runtime, tokens, or sample efficiency)?

---

> ### Author Response · Authors · 2025-11-21
>
> > W1. The main contribution, introducing turn-level rewards into PPO/GRPO, is conceptually straightforward and closely related to prior work on process reward models (PRM) and segment-level credit assignment. The paper overstates its originality by claiming to be the “first systematic study” without adequately discussing/comparing with these prior methods.
>
>
> **Response**: We thank the reviewer for raising this point. While turn-level supervision may appear conceptually straightforward in hindsight, we emphasize that incorporating explicit turn-level rewards into PPO/GRPO is not trivial in the context of multi-turn LLM agents. Most existing multi-turn agent studies rely on trajectory-level rewards, either through (i) terminal rewards that provide only a final outcome signal, or (ii) delayed rewards that merge intermediate and outcome signals into a single sparse feedback. Both design choices lead to poor credit assignment. Moreover, extending GRPO to support fine-grained credit assignment requires nontrivial modifications to its original derivation.
>
>
> We agree that PRMs and segment-level credit assignment are closely related to our work, as they also aim to provide fine-grained supervision. Turn-level rewards in our framework can indeed be instantiated using PRMs or other verifiable intermediate signals to achieve fine-grained credit assignment. While PRMs provide strong supervision and have also been extensively explored for inference-time scaling \[1,2\], they remain less explored in search-agent settings because (i) hosting an additional PRM model during training substantially increases system complexity, and (ii) training a high-quality PRM is itself nontrivial. In addition, segment-level credit assignment methods \[3,4,5,6\] are less explored in agent tasks. We have expanded the Related Work section (Section B) to more thoroughly discuss prior methods on PRMs and segment-level credit assignment。
>
>
> We also acknowledge that the original phrase “first systematic study” is an overstatement. We have revised this claim to a more moderate and accurate description in Abstract and Section 1.
> Moreover, reviewer CsNb explicitly noted that the paper provides a detailed and practical framework for designing turn-level rewards, which they recognized as a significant contribution. We respectfully ask the reviewer to consider this perspective, together with our discussion above, and to reassess the contributions and novelty of our work.
>
>
> \[1\] Lightman et al., _Let’s verify step by step_, ICLR, 2023
> \[2\] Uesato et al., _Solving math word problems with process-and outcome-based feedback_, 2022
> \[3\] Cui et al., _Process reinforcement through implicit rewards_, 2025
> \[4\] Cheng et al., _Stop summation: Min-form credit assignment is all process reward model needs for reasoning_, 2025
> \[5\] Feng et al., _Group-in-group policy optimization for llm agent training_, 2025
> \[6\] Guo et al., _Segment policy optimization: Effective segment-level credit assignment in rl for large language models_, 2025.

---

> > ### Author Response · Authors · 2025-11-21
> >
> > > W2. The experiments are limited to search-based QA tasks, leaving it unclear whether the proposed framework generalizes to other multi-turn or open-ended domains such as code generation, dialogue, or planning.
> >
> > **Response**: Thank the reviewer for this insightful question. Although our primary experiments focus on search-based QA tasks, we agree that it is important to evaluate whether our framework generalizes to other multi-turn or open-ended domains.
> >
> > To address this, we have added an additional case study in the revised manuscript using Sokoban, a multi-turn puzzle environment that is fundamentally different from search-based QA. Unlike retrieval-style tasks with well-defined turn boundaries, Sokoban requires symbolic planning, exhibits irreversible and unpredictable state transitions, and is partially observable from the perspective of an LLM agent. These characteristics make it closer in nature to open-ended planning and interactive decision-making tasks than to structured QA.
> >
> > As shown in Section E and the table below, MT-PPO consistently outperforms all PPO and GRPO baselines in both success rate and effective action rate. This demonstrates that explicit turn-level rewards and fine-grained credit assignment remains effective even in settings that involve complex multi-step reasoning, non-deterministic dynamics, and limited observability. Our results on Sokoban provide concrete evidence that the framework is not restricted to search-based QA and can extend to broader multi-turn environments that require planning and reasoning. We view applying the framework to domains such as dialogue and collaborative planning as promising future work.
> >
> > | Method     | Success Rate | Effective Action Rate |
> > |------------|--------------|------------------------|
> > | **GRPO-OR**| 0.078125     | 0.26042                |
> > | **GRPO-MR**| 0.25781      | 0.67708                |
> > | **PPO-OR** | 0.38281      | 0.69531                |
> > | **PPO-MR** | 0.50781      | 0.84505                |
> > | **MT-PPO** | **0.65625**  | **0.98698**            |
> >
> > We hope this clarifies the generality of our approach.
> > > W3. The reported improvement in answer accuracy (approximately +1.5% over PPO) is relatively modest compared to the additional implementation complexity required for designing and tuning intermediate rewards.
> >
> > **Response**: We thank the reviewer for raising this question. We respectfully note that an improvement of approximately +1.5% answer accuracy over PPO is not modest in the context of multi-turn reasoning and search-based LLM agents, where gains are typically incremental and difficult to obtain due to the high performance of strong baselines. To the best of our knowledge, our results represent the strongest performance within the Search-R1 pipeline using the original prompt.
> >
> > Moreover, beyond accuracy, our approach provides substantial benefits in training dynamics, including greater stability and faster convergence, as shown in Figure 3. In the revised manuscript, we further benchmark our method against several recent state-of-the-art RL approaches tailored for search tasks (see Table 2). Our method achieves the best overall performance across accuracy, and format correctness. These results highlight the practical value of using explicit turn-level rewards and improved credit assignment. We hope this clarification addresses the reviewer’s concern.
> >
> > > W4. The motivation for introducing MT-GRPO is weakly justified, and its evaluation is minimal.
> >
> > **Response**: We thank the reviewer for raising this concern. MT-GRPO is not our main algorithm and is just used to illustrate the importance of credit assignment. The motivation for introducing MT-GRPO is not weakly justified. In Section E, we provide a detailed derivation of MT-GRPO for the general multi-turn setting. However, due to its exponential computational cost and fixed-turn constraints, MT-GRPO is not scalable to broader agentic scenarios. This motivates our introduction of the PPO-based method with turn-level rewards, which offers a more flexible, scalable, and efficient solution.
> >
> > The evaluation of MT-GRPO is not minimal. we provide extensive comparisons. As shown in Figure 1 of the main paper and Figures 9–12 and Table 5 in Appendix F, MT-GRPO consistently outperforms vanilla GRPO, demonstrating the benefits of incorporating explicit turn-level rewards even within the GRPO framework. We hope this addresses the reviewer's concern.

---

> > > ### Author Response · Authors · 2025-11-21
> > >
> > > > W5. Novelty concern: Much of the paper’s content overlaps with existing work, and the contributions are largely incremental.
> > >
> > > **Response**: We thank the reviewer for raising this concern. We respectfully note that much of our paper’s content does not overlap with existing work, and we have revised the manuscript to more clearly highlight the novelty and contributions.
> > >
> > > First, the discussion of different MDP formulations (Section 2) and the accompanying theoretical analysis (Section C) are new. Most existing multi-turn LLM agent methods adopt MDP formulations with trajectory-level rewards, either (i) terminal rewards that provide only a final outcome signal or (ii) delayed rewards that merge intermediate and outcome signals into a single sparse feedback. These design choices lead to poor credit assignment. To address this limitation, we reformulate these tasks as MDPs with explicit turn-level rewards and provide theoretical results showing the benefits of this formulation. This perspective is, to our knowledge, absent in prior work.
> > >
> > > Second, incorporating explicit turn-level rewards into PPO/GRPO is not trivial. The derivation and analysis of MT-GRPO (Sections 3.2 and E) are new contributions. To the best of our knowledge, prior work has not provided a detailed treatment of how to adapt GRPO to support fine-grained credit assignment as thoroughly as we do.
> > >
> > > Third, we investigate the training dynamics of different RL algorithms under multi-turn settings. This aspect has received limited attention in prior work, yet is critical for understanding stability, convergence, and practical applicability. Our empirical analysis in Section 6 provides new insights into how different reward structures and RL algorithms behave in multi-turn agent training.
> > >
> > > Moreover, reviewer CsNb explicitly noted that the paper provides a detailed and practical framework for designing turn-level rewards, which they recognized as a significant contribution. We respectfully ask the reviewer to consider this perspective, together with our discussion above, and to reassess the contributions and novelty of our work.
> > >
> > > > Q1. How does MT-PPO performance compare to prior PRM or step-level RL work?
> > >
> > > **Response**: We thank the reviewer for raising this question. While PRMs provide strong supervision signals, they are less explored in search-agent tasks because hosting an additional model during training substantially increases system complexity, and training high-quality PRMs is itself nontrivial.
> > >
> > > To provide a stronger empirical comparison, we include step-level RL work \[7\]. StepSeatch trains Search-R1 using PPO with turn-level rewards based on information gain and redundancy penalty. We also include recent state-of-the-art RL baseline \[8\]. OTC trains Search-R1 using GRPO and PPO with trajectory-level rewards jointly consider correctness and tool efficiency.  The table below, along with detailed results in Table 2 and Figure 3 of Section 6, shows that our MT-PPO method achieves the best overall performance across all evaluated approaches. We hope this addresses the reviewer's concern.
> > >
> > >
> > >
> > > | Methods                 | NQ    | TriviaQA | PopQA | HotpotQA | 2wiki | Musique | Avg.  |
> > > |-------------------------|-------|----------|-------|----------|-------|---------|-------|
> > > | **GRPO (OTC)**          | 0.444 | 0.597    | 0.431 | 0.366    | 0.311 | 0.130   | 0.380 |
> > > | **PPO (OTC)**           | 0.446 | 0.623    | 0.425 | 0.383    | 0.363 | 0.152   | 0.399 |
> > > | **PPO (StepSearch)**    | 0.355 | 0.570    | 0.385 | 0.351    | 0.396 | 0.179   | 0.373 |
> > > | **MT-PPO (ours)**       | **0.490** | **0.647** | **0.459** | **0.453** | **0.424** | **0.209** | **0.447** |
> > >
> > > \[7\] Ziliang Wang, Xuhui Zheng, Kang An, Cijun Ouyang, Jialu Cai, Yuhang Wang, and Yichao Wu. _StepSearch: Igniting LLMs’ search ability via step-wise proximal policy optimization._ arXiv:2505.15107, 2025.
> > > \[8\] Hongru Wang, Cheng Qian, Wanjun Zhong, Xiusi Chen, Jiahao Qiu, Shijue Huang, Bowen Jin, Mengdi Wang, Kam-Fai Wong, and Heng Ji. _OTC: Optimal tool calls via reinforcement learning._ arXiv:2504.14870, 2025.

---

> > > > ### Author Response · Authors · 2025-11-21
> > > >
> > > > > Q2. How sensitive are results to the chosen reward weights (retrieval, format, search penalty)?
> > > >
> > > > **Response**: We thank the reviewer for raising this question. Reward design is indeed crucial in RL, and as in most RL settings, the numerical values of reward components require tuning to match different tasks and optimization objectives. While a full exploration of all reward weights is beyond our time budget and compute resource, we performed an extended ablation on the search-penalty reward to more clearly illustrate sensitivity.
> > > >
> > > > In the revised manuscript, we analyze how the search-count reward influences training dynamics and final performance. As shown in Figure 4, incorporating a moderate search-count penalty (e.g., $\lambda_s = 0.1$) significantly improves both training stability and answer correctness. The left panel indicates that $\lambda_s = 0.1$ yields the highest and most consistent accuracy, whereas stronger penalties (e.g., $\lambda_s = 0.3$) suppress useful search behavior and lead to worse outcomes.
> > > >
> > > > The middle panel further illustrates how this reward shapes the agent’s turn usage. With $\lambda_s = 0.1$, the agent quickly reduces unnecessary search calls and stabilizes around an efficient interaction pattern. In contrast, removing the penalty ($\lambda_s = 0.0$) results in unstable dynamics characterized by excessive or erratic search behavior, which ultimately impedes convergence.
> > > >
> > > >
> > > > We hope this directly addresses the reviewer’s concern.
> > > >
> > > >
> > > >
> > > >
> > > > > Q3. What is the computational overhead of MT-PPO relative to standard PPO (in terms of runtime, tokens, or sample efficiency)?
> > > >
> > > > **Response**: We thank the reviewer for raising this question. MT-PPO introduces negligible computational overhead compared to standard PPO.
> > > >
> > > > The only difference is that MT-PPO computes additional turn-level rewards during post-processing. Standard PPO computes outcome rewards once per trajectory; MT-PPO additionally computes a reward at the end of each turn and assigns it to the final token of that turn. This computation is lightweight, requires no extra model forward passes, does not increase token generation, and does not change the number of collected samples. All reward computation happens post-hoc and is inexpensive. As a result, MT-PPO has essentially the same runtime, token cost, and sample efficiency as standard PPO. We hope this directly addresses the reviewer’s concern.

---

### Official Review · Reviewer_CsNb · 2025-10-30

**Soundness:** 3
**Presentation:** 3
**Contribution:** 2
**Rating:** 6
**Confidence:** 4

**Summary:**

This paper addresses the challenge of training LLM agents for complex, multi-turn tasks using RL, proposing a novel turn-level reward design strategy. This strategy provides fine-grained credit assignment by rewarding the agent at each step of its multi-turn interaction. The authors extend both GRPO and PPO into multi-turn variants and conduct case studies on a reasoning-augmented search agent. Experiments demonstrate that their method achieves greater training stability, faster convergence, and higher accuracy compared to baseline methods that use only outcome-level rewards.

**Strengths:**

- 1.The work identifies and systematically tackles a fundamental flaw in applying RL to multi-turn LLM agents: the credit assignment problem. By shifting from sparse, end-of-task rewards to dense, turn-level rewards, the method provides the agent with much richer and more immediate feedback, which is crucial for learning complex sequences of actions.

- 2.The paper offers a detailed and practical framework for designing turn-level rewards, which is a significant contribution. It introduces two distinct types of rewards: verifiable rewards and LLM-as-judge rewards. This dual approach ensures both precision and flexibility in guiding the agent's behavior. The authors not only create a multi-turn variant of GRPO but also develop MT-PPO to overcome MT-GRPO's computational limitations.

- 3.Experiments on multiple question-answering datasets consistently show that their approach leads to more stable training, faster convergence, and superior performance in both answer correctness and output format adherence compared to strong baselines.

**Weaknesses:**

- 1. **High Computational Complexity**: The proposed MT-GRPO method requires exponential trajectory samples, making it infeasible for long-horizon tasks. While MT-PPO reduces this cost via a critic model, it still introduces additional training overhead.

- 2. **Fixed-Turn Constraint Limits Flexibility**: MT-GRPO mandates all rollout groups to have the same number of turns, enforced through system prompts. This rigid structure hinders adaptability to dynamic scenarios where tasks may require variable interaction lengths.

- 3. **Reward Design Relies on Heuristic Priors**: Turn-level rewards are manually tuned without theoretical justification. This risks reward hacking and limits generalizability. The LLM-as-Judge approach also inherits biases from the judge model.

**Questions:**

- 1. Is there a more reliable and theoretically-grounded method for reward design that enables adaptation to different tasks?
- 2. The experiments focus on structured search tasks with clear turn boundaries. How would the method perform in less structured environments, such as open-ended dialogue or collaborative planning, where turns may involve unpredictable state transitions or partial observability?

---

> ### Author Response · Authors · 2025-11-21
>
> > Q1. Is there a more reliable and theoretically-grounded method for reward design that enables adaptation to different tasks?
>
>
> **Response**: Thank the reviewer for this insightful question. In principle, a more theoretically grounded approach to reward design would be to learn an optimal reward model to enable adaptation to different tasks. However, training a reliable reward model remains an open problem in practice: it requires large-scale high-quality supervision, stable off-policy estimation, and reliable value-function learning. In contrast, turn-level verifiable rewards are **naturally** available in multi-turn agent tasks, as each complete round of agent–environment interaction yields meaningful signals that can be reliably extracted and used for RL training to improve the LLM agent’s performance, without the overhead of learning a reward model.
>
>
> In our case study on search agents, the rewards, including retrievel correctness, format correctness, search count penalty, and answer correctness, are straightforward to obtain and widely adopted in prior work \[1,2\]. We also do one more case study on multi-turn game agents in Section E, focusing on the classic grid-based puzzle Sokoban. In this environment, the agent must push boxes to designated target locations on a 2D grid using discrete actions (up, down, left, right). The reward structure arises naturally from the game dynamics, including signals such as successful box placement, distance-based progress, and turn-level format rewards. These clear intermediate signals demonstrate that turn-level rewards can be defined naturally beyond language-based tasks.
>
>
> More broadly, in scenarios where explicit rewards are unavailable, one could incorporate confidence-based signals \[3\] or learn internal rewards \[4\]. We view these directions as complementary to our method and leave them for future work. We hope this addresses the reviewer’s concern.
>
> \[1\] Jin et al.,  _Search-R1: Training LLMs to reason and leverage search engines with reinforcement learning_, 2025.
> \[2\] Jin et al., _An empirical study on reinforcement learning for reasoning–search interleaved LLM agents_, 2025.
> \[3\] He et al., _Beyond Correctness: Confidence-Aware Reward Modeling for Enhancing Large Language Model Reasoning_, 2025.
> \[4\] Zhao et al., _Learning to reason without external rewards_, 2025.
> \[5\] Schrader, _gym-sokoban_, 2018
>
> > Q2. The experiments focus on structured search tasks with clear turn boundaries. How would the method perform in less structured environments, such as open-ended dialogue or collaborative planning, where turns may involve unpredictable state transitions or partial observability?
>
> **Response**: We thank the reviewer for raising this point. We agree that our experiments focus on structured search tasks with clear turn boundaries. To evaluate our method beyond structured search tasks, we conducted an additional case study on a less structured and more challenging environment, where turns may involve unpredictable state transitions and partial observability.
>
>
> In the revised manuscript, we present an additional case study on multi-turn game agents, focusing on the classic grid-based puzzle Sokoban \[5\]. In this puzzle, the agent must push all boxes to designated target locations. The environment is represented as a 2D grid, and the action space is discrete (up, down, left, right). The key challenge is that Sokoban is irreversible: boxes can be pushed but not pulled, so a single wrong move may lead to an unrecoverable dead-end. This requires multi-step symbolic planning and introduces unpredictable state transitions, as pushing a box may dramatically change the future solvability of the puzzle. Furthermore, Sokoban is partially observable from the perspective of an LLM-based agent, which must infer spatial dynamics from visual descriptions rather than an explicit state vector.
>
>
> In Section E, our experimental results show that MT-PPO consistently outperforms all baseline methods (PPO-MR, PPO-OR, GRPO-OR, GRPO-MR) in both success rate and effective action rate. These findings further demonstrate that explicit turn-level rewards and fine-grained credit assignment remain beneficial even in environments that are less structured and harder to model.
>
> We hope this addresses the reviewer’s concern and helps clarify the generality of our approach beyond structured search tasks.

---

### Author Response · Authors · 2025-11-21
**Summary of paper revision**

Dear reviewers and AC,

We thank all reviewers for their valuable comments. In response to the comments, we have revised the original submission. The main updates are summarized below:

1.  **New theoretical support** (Reviewer `JWKY`): We revised **Section 2** to more clearly compare different MDP formulations and added new theoretical analysis in **Section C** to better justify the advantage of the turn-level MDP with explicit turn-level rewards adopted in this paper.
2.  **Expanded baseline comparisons and ablation studies** (Reviewers `SxgQ` and `JWKY`): We incorporated several additional state-of-the-art RL baselines for search agents, including PPO-MR, OTC, and StepSearch. The corresponding results are reported in **Section 6.3**. We also added additional ablation studies on reward design in **Section 6.4**.
3.  **Additional application and evaluation** (Reviewers `CsNb` and `SxgQ`): We extended our experiments to a game-agent environment (Sokoban) to demonstrate the generality of our approach beyond structured search tasks in **Section E**.
4.  **Broader discussion of related work** (Reviewers `SxgQ` and `JWKY`): We expanded the Related Work section (**Section B**) to more thoroughly discuss PRMs, segment-level credit assignment, and other concurrent methods.


Best regards,
Authors of Submission 16065

---

### Author Response · Authors · 2025-12-03
**Rebuttal Summary**

Dear Reviewers, ACs, SACs and PCs,

We sincerely thank you for your time, efforts, and insightful evaluation of our manuscript. Unfortunately, due to an unexpected issue in this year’s ICLR discussion phase, the reviewers were unable to participate in further discussion after we submitted our detailed responses. We regret losing the opportunity to clarify our work and address any remaining questions. Nevertheless, we want to emphasize that our rebuttal provides thorough explanations and comprehensive evidence, and we firmly believe that it fully resolves the concerns raised in the initial reviews.



To help facilitate a quick understanding of our work and our responses, we provide a concise summary below.


**Key strengths highlighted by reviewers**

- [Reviewers `SxgQ`, `CsNb`] **Addresses a core challenge in multi-turn LLM agents.** Reviewers emphasized that the paper tackles a central problem in multi-turn RL, particularly the difficulty of credit assignment and reward shaping. They noted that focusing on this foundational issue is both timely and impactful.
- [Reviewers `JWKY`, `SxgQ`, `CsNb`] **Introduces a practical and effective turn-level reward framework.** The proposed turn-level reward design was recognized as a well-motivated and useful contribution for guiding multi-turn agent learning. Reviewers highlighted that it offers a clear and structured way to incorporate step-level feedback.
- [Reviewers `JWKY`, `CsNb`] **Demonstrates consistent empirical improvements.** Reviewers pointed to notable gains in training stability, convergence behavior, and overall performance compared to standard baselines. They considered these improvements strong evidence of the method’s effectiveness.
- [Reviewers `SxgQ`, `CsNb`] **Presents the ideas clearly and effectively.** The paper was praised for its coherent presentation, clear writing, and illustrative examples. Reviewers found the exposition accessible and helpful for understanding the key contributions.



**Key responses to each reviewers.**
- **Reviewer CsNb:**  We address the questions on reward design and generality by clarifying that turn-level verifiable rewards arise naturally in multi-turn agent tasks. We added a new non-structured environment (Sokoban) and show that MT-PPO consistently outperforms all baselines, demonstrating robustness beyond search QA.
- **Reviewer SxgQ:**  We clarify the novelty: integrating explicit turn-level rewards into PPO/GRPO requires nontrivial modifications, and we expanded Related Work to better differentiate from PRMs and segment-level methods. We added new baselines (StepSearch, OTC), new analyses, revised claims, and provided Sokoban experiments to show generalization.
-  **Reviewer JWKY:** We strengthened the theoretical component with new variance analysis comparing MDP formulations and added several state-of-the-art RL baselines. We also discussed concurrent work and clarified the distinctions. MT-PPO achieves the strongest performance across all benchmarks.


**Key revisions in response to the comments**
1.  **New theoretical support** (Reviewer `JWKY`): We revised **Section 2** to more clearly compare different MDP formulations and added new theoretical analysis in **Section C** to better justify the advantage of the turn-level MDP with explicit turn-level rewards adopted in this paper.
2.  **Expanded baseline comparisons and ablation studies** (Reviewers `SxgQ` and `JWKY`): We incorporated several additional state-of-the-art RL baselines for search agents, including PPO-MR, OTC, and StepSearch. The corresponding results are reported in **Section 6.3**. We also added additional ablation studies on reward design in **Section 6.4**.
3.  **Additional application and evaluation** (Reviewers `CsNb` and `SxgQ`): We extended our experiments to a game-agent environment (Sokoban) to demonstrate the generality of our approach beyond structured search tasks in **Section E**.
4.  **Broader discussion of related work** (Reviewers `SxgQ` and `JWKY`): We expanded the Related Work section (**Section B**) to more thoroughly discuss PRMs, segment-level credit assignment, and other concurrent methods.
5.  **Other clarifications and questions**:  We provide detailed analysis and explanation as requested by each reviewers.



We sincerely appreciate the reviewers’ constructive feedback and would like to highlight that our rebuttal directly and comprehensively addresses all raised concerns. We respectfully ask the ACs, SACs, and PCs to consider our full responses and **recognize that the reviewers’ perspectives might have shifted had the discussion phase not been unexpectedly interrupted**. Although we regret the lost opportunity for further clarification, we fully respect the review process and conference guidelines.

Thank you again for your time and careful consideration. We are grateful for your efforts and hope this summary is useful.



Best regards,
Authors of Submission 16065

---

### Meta-Review · Area_Chair_FDyY · 2026-01-02

**Summary:**

This paper introduced a turn-level reward design strategy to enhance RL for LLM agents in complex, long-horizon tasks. Addressing the limitations of trajectory level rewards, the authors reformulate multi-turn tasks as MDP with explicit intermediate rewards. With extending popular algorithms into multi-turn variants (MT-GRPO and MT-PPO) and evaluating them on diverse tasks, they yields significant improvements in training stability, convergence, and overall accuracy.

**Reviewer Concerns:**

Reviewers raised diverse concerns on this paper. The main concern comes from the novelty side as intermediate reward is not a novel idea and there are various previous works that shared the same idea. Some reviewers also raised the issues on the computational complexity. They also have some concerns on limited experiments with limited baselines. Although the authors responded to those reviewers' concerns, some of the responses seem not enough especially on the novelty, computational complexities, as well as significant expansions of the experiments.

**Reviewer Scores:**

Three reviewers provided 6, 4, 2 scores initially. Unfortunately, no reviewers participated into the discussion. I went over all the reviews as well as the response and make this decision.

---

### Decision · Program_Chairs · 2026-01-26

Reject